# SOLD: Slot Object-Centric Latent Dynamics Models for Relational Manipulation Learning from Pixels

Malte Mosbach [* 1]   Jan Niklas Ewertz [* 2 3]   Angel Villar-Corrales [1]   Sven Behnke [1]

## Abstract

Learning a latent dynamics model provides a task-agnostic representation of an agent's understanding of its environment. Leveraging this knowledge for model-based reinforcement learning (RL) holds the potential to improve sample efficiency over model-free methods by learning from imagined rollouts. Furthermore, because the latent space serves as input to behavior models, the informative representations learned by the world model facilitate efficient learning of desired skills. Most existing methods rely on holistic representations of the environment's state. In contrast, humans reason about objects and their interactions, predicting how actions will affect specific parts of their surroundings. Inspired by this, we propose *Slot-Attention for Object-centric Latent Dynamics (SOLD)*, a novel model-based RL algorithm that learns object-centric dynamics models in an unsupervised manner from pixel inputs. We demonstrate that the structured latent space not only improves model interpretability but also provides a valuable input space for behavior models to reason over. Our results show that SOLD outperforms DreamerV3 and TD-MPC2 – state-of-the-art model-based RL algorithms – across a range of multi-object manipulation environments that require both relational reasoning and dexterous control. Videos and code are available at https://slot-latent-dynamics.github.io.

[*]Equal contribution [1]Autonomous Intelligent Systems Group, Computer Science Institute VI – Intelligent Systems and Robotics, Center for Robotics, and the Lamarr Institute for Machine Learning and Artificial Intelligence, University of Bonn, Germany [2]Intelligent Robot Perception Group, Informatics Institute for Anthropomatics and Robotics, Karlsruhe Institute of Technology, Germany [3]Mercedes-Benz AG, Sindelfingen, Germany. Correspondence to: Malte Mosbach <mosbach@ais.uni-bonn.de>.

*Proceedings of the 42$^{nd}$ International Conference on Machine Learning*, Vancouver, Canada. PMLR 267, 2025. Copyright 2025 by the author(s).

## 1. Introduction

Advances in reinforcement learning (RL) have showcased the ability to learn sophisticated control strategies through interaction, achieving superhuman performance in domains ranging from board games (Silver et al., 2016) to drone racing (Kaufmann et al., 2023). While these approaches excel when explicit models of the environment are available or abundant data can be collected, learning complex control tasks in a sample-efficient manner remains a significant challenge. *Model-based RL* (MBRL) has emerged as a promising approach to address this limitation by constructing models of the environment dynamics. For instance, the Dreamer framework (Hafner et al., 2019; 2020; 2025) improves sample efficiency over model-free methods by learning behaviors via imagined rollouts.

While these research efforts have produced world models capable of accurately predicting the dynamics of visual tasks, they rely on a *holistic representation* of the environment state. In contrast, humans perceive the world by parsing scenes into individual objects (Spelke, 1990), anticipating how their actions will influence specific components of their surroundings. Relational reasoning, particularly in environments with multiple interacting objects, is a cornerstone of human intelligence and a crucial capability for robotic manipulation tasks (Battaglia et al., 2018). Introducing *structured, object-centric representations* into MBRL represents a powerful inductive bias, enabling agents to reason about task-relevant components of the environment while ignoring irrelevant details. Such structured representations not only enhance interpretability but also improve the efficiency of behavior learning by providing models with meaningful latent spaces. Despite these advantages, the integration of object-centric representations and world models remains largely underexplored. To the best of our knowledge, no prior method learns purely inside imagined rollouts of an object-centric world model trained from pixel inputs.

To address the limitations of holistic representations in MBRL, we propose *SOLD*, a novel algorithm that leverages structured, object-centric states within the latent space of its world model. The contributions of our method are twofold. First, we introduce an *object-centric dynamics model* that predicts future frames in terms of their slot representation.

Building on the OCVP framework (Villar-Corrales et al., 2023), we introduce an action-conditional dynamics model, enabling accurate forecasting of action effects on individual objects. Notably, the dynamics model is trained solely from pixels through a loss on the reconstructions and slot representations of the predicted frames, bypassing the need for object annotations and facilitating scalability to complex visual tasks. Second, we propose the *Slot Aggregation Transformer*, a novel architectural backbone that aggregates information from the history of object slots to make reward, value, and action predictions. This enables efficient MBRL training grounded in structured representations.

For systematic evaluation, we introduce a suite of visual robotics tasks, shown in Figure 1, that require varying levels of relational reasoning and manipulation capabilities. We perform an extensive comparison on this benchmark, demonstrating that our method achieves superior performance to both state-of-the-art MBRL algorithms DreamerV3 (Hafner et al., 2025) and TD-MPC2 (Hansen et al., 2024). Further, to highlight its broader potential, we apply SOLD to tasks from two RL benchmarks that are not object-centric by design, providing evidence of the generalizability of our framework. In summary, we make the following contributions:

- We introduce SOLD, the first MBRL algorithm to learn inside imagined rollouts of an object-centric dynamics model trained directly from pixel inputs, achieving state-of-the-art performance on visual robotics tasks that require both relational reasoning and manipulation.

- By visualizing learned attention weights, we show that our method produces human-interpretable attention patterns, providing insights into the decision-making process of behavior models.

- We overcome limitations of prior object-centric RL methods by showing that our encoder-decoder module can (i) be adapted to state distributions vastly different from those seen under random pre-training, and (ii) generalize to environments that are not object-centric by design.

## 2. Background

**Slot Attention for Video (SAVi)**   SOLD employs SAVi (Kipf et al., 2022), an encoder-decoder architecture with a structured bottleneck composed of $N$ permutation-equivariant object embeddings, referred to as slots. It recursively parses a sequence of video frames $\boldsymbol{o}_{0:\tau} = \boldsymbol{o}_0, ..., \boldsymbol{o}_\tau$ into their object representations $\boldsymbol{Z}_{0:\tau} = \boldsymbol{Z}_0, ..., \boldsymbol{Z}_\tau, \boldsymbol{Z}_t \in \mathbb{R}^{N \times D_Z}$. At time $t$, SAVi encodes the input video frame $\boldsymbol{o}_t$ into a set of feature maps $\boldsymbol{F}_t \in \mathbb{R}^{L \times D_h}$, where $L$ is the size of the flattened grid (i.e. $L = \text{width} \cdot \text{height}$), and uses Slot Attention (Locatello et al., 2020) to iteratively refine

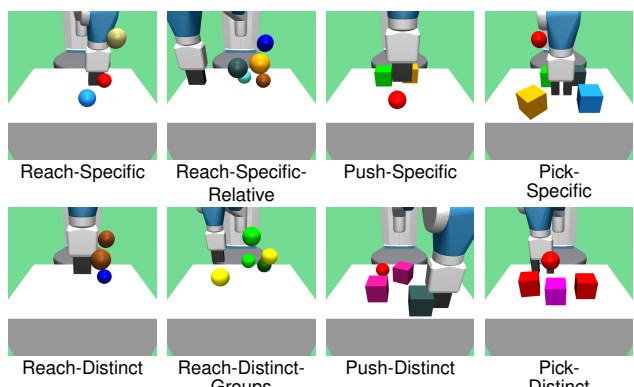

Figure 1: Suite of visual environments requiring relational reasoning and low-level manipulation to be solved.

the previous slot representations conditioned on the current features. Slot Attention performs cross-attention between the slots and image features with the attention coefficients normalized over the slot dimension, thus encouraging the slots to compete to represent feature locations:

$$\boldsymbol{A} \doteq \operatorname*{softmax}_{N}\left(\frac{q(\boldsymbol{Z}_{t-1}) \cdot k(\boldsymbol{F}_t)^T}{\sqrt{D}}\right) \in \mathbb{R}^{N \times L}, \quad (1)$$

where $q(.)$ and $k(.)$ are learned linear mappings to a common dimension $D$. The slots are then independently updated via a shared Gated Recurrent Unit (Cho et al., 2014) (GRU) followed by a residual MLP:

$$\boldsymbol{Z}_t \doteq \text{MLP}(\text{GRU}(\boldsymbol{A} \cdot v(\boldsymbol{F}_t), \boldsymbol{Z}_{t-1})),$$
$$\boldsymbol{A}_{n,l} \doteq \frac{\boldsymbol{A}_{n,l}}{\sum_{i=0}^{L-1} \boldsymbol{A}_{n,i}}, \quad (2)$$

and $v(.)$ is a learned linear projection. The steps described in Equations 1 and 2 can be repeated multiple times with shared weights to iteratively refine the slots and obtain an accurate object-centric representation of the scene.
Finally, SAVi independently decodes each slot of $\boldsymbol{Z}_t$ into per-object images and alpha masks, which can be normalized and combined via weighted sum to render video frames. SAVi is trained end-to-end in a self-supervised manner with an image reconstruction loss.

**Object-Centric Video Prediction (OCVP)**   Our dynamics model builds on OCVP (Villar-Corrales et al., 2023) in order to autoregressively predict future object slots conditioned on past object states. OCVP is a transformer-encoder model (Vaswani et al., 2017) that decouples the processing of object dynamics and interactions, thus leading to interpretable and temporally consistent object predictions while retaining the inherent permutation-equivariant property of the object slots. This is achieved through the use of two specialized self-attention variants: *temporal attention* updates a slot representation by aggregating information from

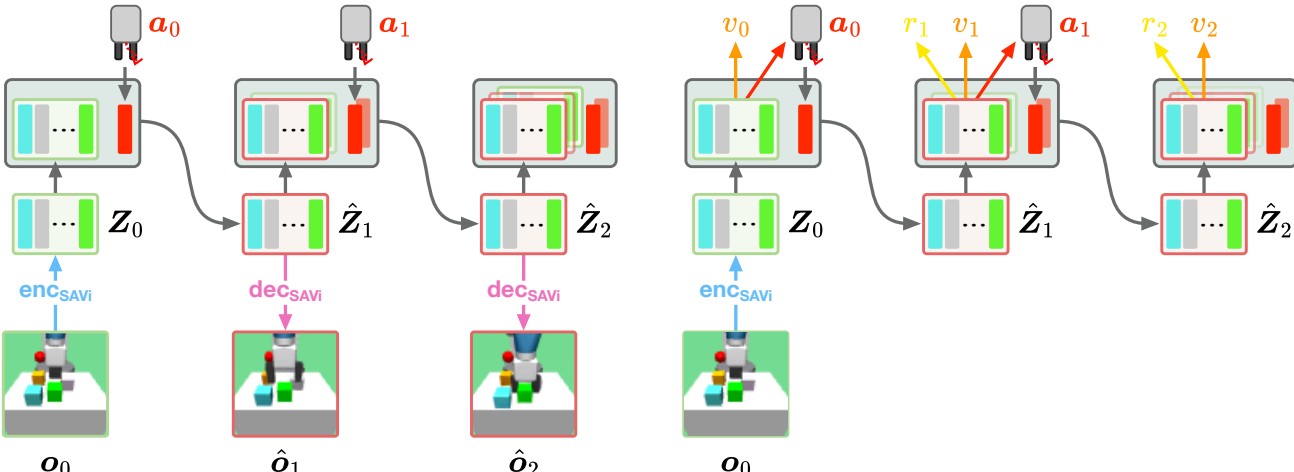

(a) World Model Learning: SAVi encodes images $o_t$ into slots $Z_t$, which are predicted by the dynamics model given the history of slots and actions. We reconstruct the images and compute their actual slot representation to shape the dynamics prediction.

(b) Behavior Learning: Actor and critic are trained via imagined rollouts in the latent space of the world model. Trajectories start after $S$ seed frames (visualized for $S = 1$) and predict forward with actions $a_t$ sampled from the actor network.

Figure 2: *SOLD* is trained by concurrently making the world model consistent with replayed experiences and learning behaviors through latent imagination.

the corresponding slot up to the current time step, without modeling interactions between distinct objects, whereas *relational attention* models object interactions by jointly processing all slots from the same time step.

## 3. Slot-Attention for Object-centric Latent Dynamics

We propose *SOLD*, a method that combines model-based RL with object-centric representations. The three core components of our method are: the *object-centric world model*, which predicts the effects of actions on the environment, the *critic*, which estimates the value of a given state, and the *actor* that selects actions to maximize this value.

Figure 2 gives an overview of the training process. The world model operates on structured latent states by splitting the environment into its constituent objects and then composing future frames via the predicted states of these individual components. Specifically, we pretrain a SAVi encoder-decoder model (Kipf et al., 2022) on random sequences from the environment to extract object-centric representations. After pretraining, all components of the world model are trained jointly using replayed experiences from the agent's interaction with the environment. The actor and critic are trained on imagined sequences of structured latent states. We execute actions sampled from the actor model in the environment and append the resulting experiences to the replay buffer. Detailed explanations of world model learning and behavior learning are provided in Sections 3.1 and 3.2, respectively.

### 3.1. World Model Learning

World models compress an agent's experience into a predictive model that forecasts the outcomes of potential actions. By simulating rollouts within the internal model, agents can learn desired behaviors in a sample-efficient manner. When the inputs are high-dimensional images, it is helpful to learn compact state representations, enabling prediction within this latent space. This type of model, called latent dynamics model, allows for efficient prediction of many latent sequences in parallel.

Most prior works rely on generating a single, holistic representation of the environment state, which contrasts with findings from cognitive psychology. Humans perceive scenes as compositions of objects (Spelke, 1990) and reason about how their actions affect distinct parts of their environment. Furthermore, environment dynamics can be compactly explained in terms of objects and their interactions (Battaglia et al., 2016). Therefore, we propose to structure the latent space by decomposing visual environments into their constituent parts.

**Components** To create a world model that operates on object-centric latent representations, we build on top of OCVP (Villar-Corrales et al., 2023). We begin by pretraining SAVi on a static dataset of frames collected from random episodes. Having a sufficiently large initial dataset is crucial for meaningful object-centric representations to emerge. These pretrained representations serve as the foundation for SOLD's object-centric world model. However, we do not freeze the pretrained encoder-decoder models, allowing

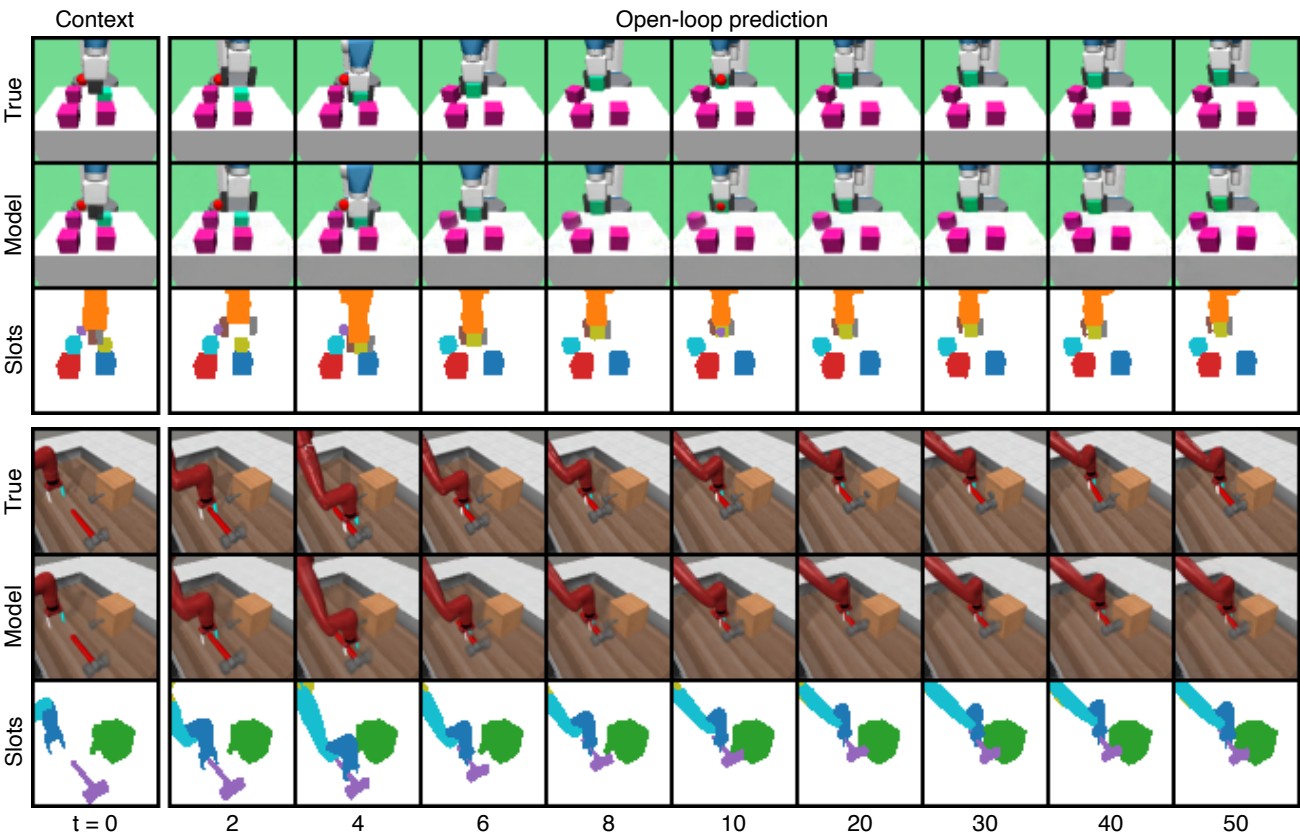

Figure 3: Open-loop predictions of our object-centric dynamics model for the *Pick-Distinct* task (top) and the *Hammer* environment from MetaWorld (bottom). Starting from a single context frame, our model predicts the next 50 frames by propagating slot representations forward without access to any intermediate images.

slots to adapt to novel configurations that do not occur during random pre-training. The sequence of object slots $\boldsymbol{Z}_{0:t}$ alongside the action commands $\boldsymbol{a}_{0:t}$ serve as inputs to our transformer-based dynamics model which predicts the slot representation of the next frame $\hat{\boldsymbol{Z}}_{t+1}$:

$$
\begin{aligned}
\text{Encoder:} \qquad & \boldsymbol{Z}_t = e_\eta(\boldsymbol{o}_t), \\
\text{Decoder:} \qquad & \hat{\boldsymbol{o}}_t = d_\eta(\boldsymbol{Z}_t), \\
\text{Dynamics model:} \qquad & \hat{\boldsymbol{Z}}_{t+1} = p_\psi(\boldsymbol{Z}_{0:t}, \boldsymbol{a}_{0:t}), \text{ and} \\
\text{Reward predictor:} \qquad & \hat{r}_t \sim p_\zeta(\hat{r}_t \mid \boldsymbol{Z}_{0:t}).
\end{aligned} \tag{3}
$$

**Object-centric Dynamics Learning**  For the dynamics model, we use the sequential attention pattern proposed by Villar-Corrales et al. (2023), which disentangles relational and temporal attention to decouple object dynamics and interactions. During training, we provide the slot representation of $S$ seed frames as context. We append the predictions to the context and apply this process in an autoregressive manner to predict the subsequent $T$ frames. We do not employ teacher forcing so that the dynamics model learns to handle its own imperfect predictions. To shape the predicted representations, we reconstruct the subsequent

frame $\hat{\boldsymbol{o}}_{t+1}$ and extract the SAVi representations of the actual frame $\boldsymbol{Z}_{t+1}$ to compute the hybrid dynamics loss:

$$
\mathcal{L}_{\text{dyn}}(\psi) \doteq \sum_{t=S}^{S+T-1} \Big[ \underbrace{\big\| \hat{\boldsymbol{Z}}_t - e_\eta(\boldsymbol{o}_t) \big\|_2^2}_{\text{Joint embedding}} + \underbrace{\big\| \hat{\boldsymbol{o}}_t - \boldsymbol{o}_t \big\|_2^2}_{\text{Reconstruction}} \Big]. \tag{4}
$$

For all loss terms, we specify the parameter group that is being optimized and omit stop-gradient notations for other models to avoid cluttering the notation.

**Reward Model Learning**  The reward predictor solves a regression problem that maps slot representations to scalar reward values, where the prediction depends on the set of slots rather than being tied to any specific one. To address this, we introduce the *Slot Aggregation Transformer (SAT)* as an architectural backbone, which introduces output tokens and a variable number of register tokens for all timesteps. Register tokens, recently shown to enhance computation in vision transformers (Darcet et al., 2024), can aid computation when processing a set of inputs to produce a singular output. To encode positional information, we adopt ALiBi (Press et al., 2022) in place of absolute position en-

coding. ALiBi introduces linear biases directly into the attention scores, effectively encoding token recency. This approach helps to generalize to sequences longer than those seen during training. A detailed description of the SAT can be found in Section B.3. To efficiently represent a wide range of reward values, we avoid directly predicting a scalar reward. Instead, the MLP head $f_\zeta$ outputs logits of a softmax distribution over $K$ exponentially spaced bins $b_i$. The predicted reward can then be computed as the expectation over these bins:

$$b \doteq \mathrm{symexp}([-20, ..., +20]),$$
$$\hat{r}_t \doteq \mathrm{softmax}(f_\zeta^{\mathrm{MLP}}(\boldsymbol{h}_t))^T \boldsymbol{b}, \qquad (5)$$

where $\boldsymbol{h}_t$ are the output tokens after being processed by the SAT backbone. To formulate the loss, the true reward $r_t$ is first transformed using the symlog function (Webber, 2012) and then encoded via a two-hot encoding strategy (Bellemare et al., 2017; Schrittwieser et al., 2020). The model is trained to maximize the log-likelihood of the two-hot encoded reward distribution under the predicted distribution:

$$\mathcal{L}_{\mathrm{rew}}(\zeta) \doteq -\sum_{t=0}^{T-1} \log p_\zeta(r_t \mid \boldsymbol{Z}_{0:t}). \qquad (6)$$

### 3.2. Behavior Learning

Our strategy of using the world model for behavior learning builds upon the Dreamer framework. At the core of this method lies the process of latent imagination, visualized in Figure 2b, which trains the actor and critic networks purely on imagined trajectories predicted by the world model. Since both the actor and critic operate on the latent state, they benefit from the structured representation learned by the world model. The architecture of both models mirrors that of the reward predictor, consisting of a SAT backbone that processes the slot histories followed by an MLP head:

$$\begin{array}{ll} \text{Actor:} & \boldsymbol{a}_t \sim \pi_\theta(\boldsymbol{a}_t \mid \boldsymbol{Z}_{0:t}), \\ \text{Critic:} & \hat{R}_t \doteq \mathrm{E}[v_\phi(\hat{R}_t \mid \boldsymbol{Z}_{0:t})]. \end{array} \qquad (7)$$

**Critic Learning** To account for rewards beyond the imagination horizon $T = 15$, the critic is trained to estimate the expected return under the current actor's behavior. Since no ground truth is available for these estimates, we compute bootstrapped $\lambda$-returns (Sutton & Barto, 2018), $R^\lambda$, via temporal difference learning. These returns integrate predicted rewards $\hat{r}$ and values $\hat{R}$ to form the target for the value model:

$$R_t^\lambda \doteq \hat{r}_{t+1} + \gamma\left((1-\lambda)\hat{R}_{t+1} + \lambda R_{t+1}^\lambda\right), \qquad (8)$$

where $R_T^\lambda \doteq \hat{R}_T$, which is trained to minimize the resulting loss:

$$\mathcal{L}_{\mathrm{critic}}(\phi) \doteq -\sum_{t=0}^{T-1} \log v_\phi(R_t^\lambda \mid \boldsymbol{Z}_{0:t}). \qquad (9)$$

Table 1: Final success rates (% ± standard deviation) of SOLD and baseline methods for the *specific* (top) and *distinct* (bottom) task variants.

(a) Specific tasks requiring mainly robotic control.

| Task | DreamerV3 | TD-MPC2 | w/o OCE | SOLD |
|---|---|---|---|---|
| Reach | 87.4 ±1 | 97.6 ±0 | 83.2 ±2 | **97.9** ±0 |
| Reach-Rel. | 45.6 ±6 | 79.1 ±1 | 39.2 ±3 | **91.1** ±2 |
| Push | **97.1** ±1 | 72.7 ±3 | 75.2 ±2 | 82.8 ±2 |
| Pick | **96.7** ±1 | 87.6 ±2 | 22.9 ±11 | 85.8 ±7 |
| **Average** | 81.7 ±21 | 84.2 ±9 | 55.1 ±25 | **89.4** ±6 |

(b) Distinct tasks requiring challenging relational reasoning.

| Task | DreamerV3 | TD-MPC2 | w/o OCE | SOLD |
|---|---|---|---|---|
| Reach | 14.6 ±6 | 31.4 ±3 | 11.3 ±1 | **91.8** ±1 |
| Reach-Gr. | 13.9 ±2 | 15.7 ±2 | 5.1 ±1 | **69.6** ±2 |
| Push | 70.0 ±5 | 12.2 ±5 | 10.5 ±1 | **80.6** ±5 |
| Pick | 33.9 ±36 | 9.8 ±1 | 0.7 ±0 | **56.4** ±25 |
| **Average** | 33.1 ±23 | 17.3 ±8 | 6.9 ±4 | **74.6** ±13 |

We decouple the gradient scale from value prediction through same approach as in the reward model, predicting a categorical distribution over exponentially spaced bins. To stabilize learning, we regularize the critic's predictions towards the outputs of an exponentially moving average (EMA) of its own parameters (Mnih et al., 2015; Hafner et al., 2025).

**Actor Learning** The actor is optimized to select actions that maximize its expected return while encouraging exploration through an entropy regularizer. Its model architecture is similar to the critic and reward predictor, but instead of regressing a scalar value, it predicts the parameters of the action distribution. Specifically, the MLP head outputs the mean $\boldsymbol{\mu}_t$ and standard deviation $\boldsymbol{\sigma}_t$ of a normal distribution $\mathcal{N}(\boldsymbol{\mu}_t, \boldsymbol{\sigma}_t | \boldsymbol{Z}_{0:t})$ over possible actions. The trade-off in the actor's loss function weights expected returns with maintaining randomness in the outputs and is hence subject to reward scale and frequency of the current environment. To adapt to varying scales of value estimates across different environments, we use a normalization factor $\mathrm{s}_V$:

$$\mathcal{L}_{\mathrm{actor}}(\theta) \doteq -\sum_{t=0}^{T-1} \frac{\hat{R}_t^\lambda}{\max(1, \mathrm{s}_V)} + \eta \mathrm{H}(\pi_\theta(\boldsymbol{a}_t \mid \boldsymbol{Z}_{0:t})), \quad (10)$$

where the value normalization is computed via the EMA of the 5th and 95th percentile of the value estimates (Hafner et al., 2025):

$$\mathrm{s}_V \doteq \mathrm{EMA}\left(\mathrm{Per}(\hat{R}_t^\lambda, 95) - \mathrm{Per}(\hat{R}_t^\lambda, 5), 0.99\right). \quad (11)$$

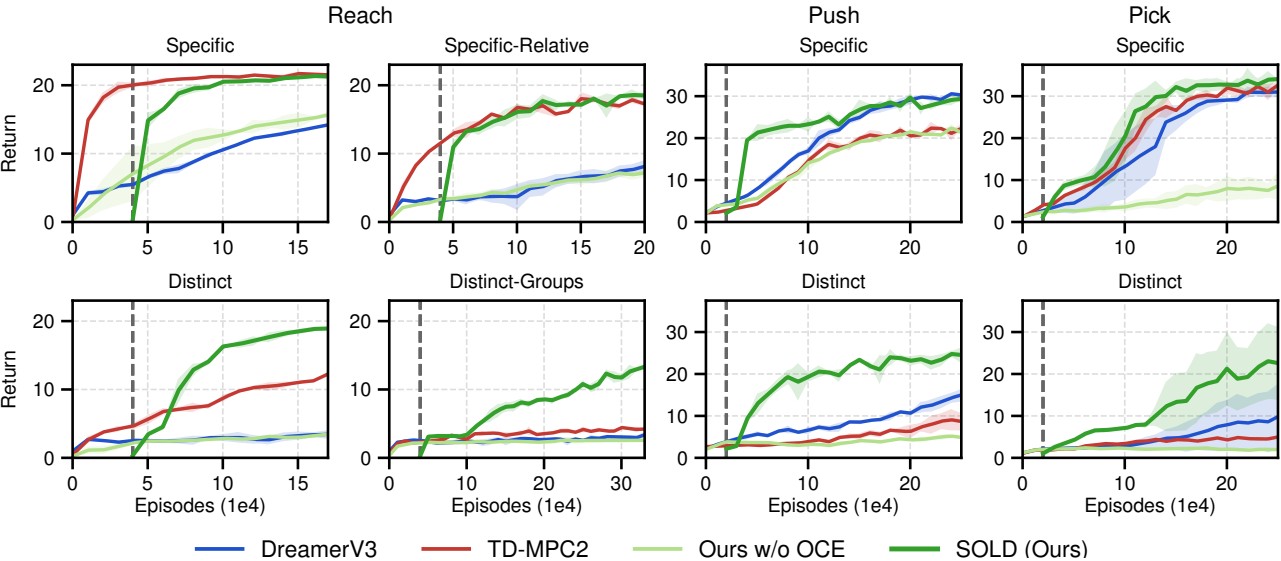

Figure 4: Achieved returns over the training duration across the eight benchmark environments. The dashed vertical line represents the offset for our method to account for the samples used during pre-training.

## 4. Results

In this section, we present the empirical evaluation of SOLD on our suite of visual continuous control tasks. We first describe the comparative baselines and the environments used in our experiments. Using this setup, we aim to answer the following questions: (a) Does SOLD accurately model object-centric dynamics in action-conditional settings, preserving the decomposition of visual scenes? (b) Does the structured latent space allow SOLD to outperform SoTA baselines on tasks that require relation reasoning capabilities? (c) Can SOLD generalize to visual environments that are not object-centric by design?

**Baselines** The chosen baselines in our evaluation serve two primary purposes: assessing the impact of the object-centric paradigm in our method and benchmarking it against state-of-the-art approaches. To evaluate the effect of object-centric representations, we compare our method to a baseline that replaces the object-centric encoder-decoder modules with a standard convolutional architecture (w/o OCE). To benchmark against the best available methods, we include DreamerV3 (Hafner et al., 2025) and TD-MPC2 (Hansen et al., 2024). Both are widely recognized for their strong performance across a wide range of tasks. Additional details for the baselines can be found in Appendix C.

**Environments** We introduce a suite of eight object-centric robotic control environments designed to test both relational reasoning and manipulation capabilities. These environments feature two types of problems: *Reach* tasks, where the agent must identify a target and move the end-effector to

its location, and manipulation tasks (*Push* and *Pick*), where the agent identifies a target block and moves it to a designated goal. To test varying levels of relational reasoning difficulty, we design the following configurations:

- *Specific* The target object is red, with 0 to 4 distractor objects of random, distinct colors present in the scene.
- *Distinct* Inspired by the odd-one-out task in cognitive science (Crutch et al., 2009; Beatty & Vartanian, 2015), this task presents 3 to 5 objects, and the target is the one that differs in color from all the others.

For the *Reach* task, we include two advanced variants:

- *Specific-Relative* The goal is to reach the reddest object, determined by the perceptual CIEDE2000 (Sharma et al., 2005) distance.
- *Distinct-Groups* The environment contains 5 targets, and the goal is to reach the one that appears only once.

On these two additional reach tasks, we reuse the SAVi models that were pre-trained for *Reach-Specific* and *Reach-Distinct*, respectively without modification. Further details about these environments are provided in Appendix D.

### 4.1. Object-centric Dynamics Learning

The object-centric representations learned by SAVi can be seen in the context frames in Figure 3. The slots effectively decompose the visual scene, with most slots representing distinct objects and three slots capturing different parts of the respective robots. This part-whole segmentation demonstrates that the slots can meaningfully identify separate parts

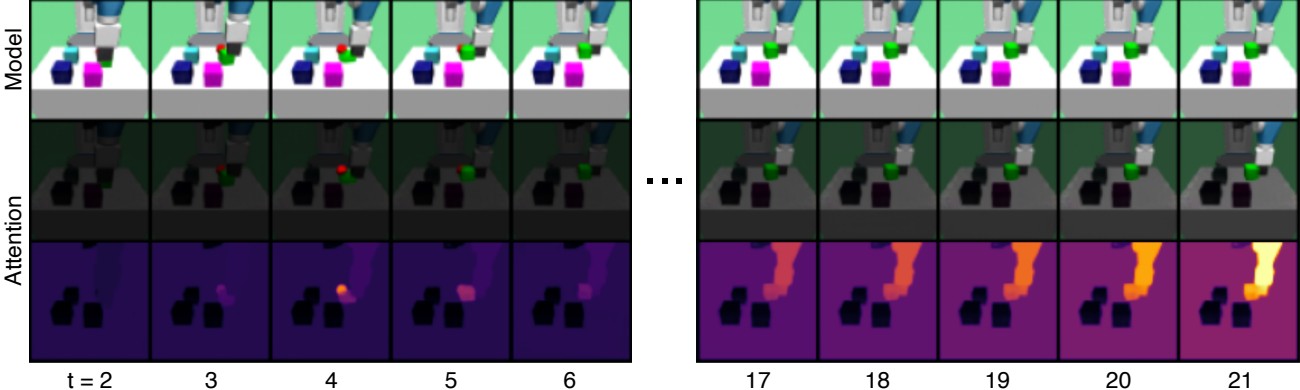

Figure 5: SOLD discovers objects relevant for task completion in an unsupervised manner over long horizons. We depict the normalized attention of the [out] token of the actor over the object tokens using Attention Rollout (Abnar & Zuidema, 2020). The full slot history is shown in Figure 16.

of a larger object, representing the gripper jaws in the first example and different parts of the kinematic chain of the Sawyer robot in the second. Notably, the sharp mask predictions show that each slot isolates information about the specific object it represents (see also Section E). This property is crucial for object-centric behavior learning, as it enables subsequent components to reason about task-relevant objects while ignoring irrelevant information. Further, the open-loop prediction of 50 future frames starting from a single seed frame, shown in Figure 3, demonstrate the model's ability to generate physically accurate predictions over long horizons. The movements of the robot and blocks are predicted with high accuracy, demonstrating the model's ability to precisely capture physical interactions between objects. Moreover, the model effectively handles occlusions, as evidenced by the continued precise prediction of the spherical red target. Importantly, the autoregressive dynamics model maintains a precise and meaningful decomposition of the scene in its predictions, even far into the future. We encourage readers to view the videos of object-centric open-loop predictions on our project page for a qualitative assessment of these results.

### 4.2. Behavior Learning

To assess SOLD's performance across our robotic control tasks, we train each method with three different random seeds per environment. The final success rates achieved by each method are presented in Table 1. SOLD consistently outperforms the non-object-centric baseline, often by a significant margin, underscoring the effectiveness of object-centric representations for the considered tasks. When compared to state-of-the-art MBRL methods, SOLD demonstrates competitive or superior performance. Notably, while SOLD narrowly surpasses DreamerV3 and TD-MPC2 across the *Specific* tasks, it significantly outperforms them

on the more challenging *Distinct* variants, which require complex relational reasoning between objects. On these tasks, SOLD achieves more than double the performance of the second-best method.

Beyond that, when examining the performance over the course of training, as shown in Figure 4, we observe additional advantages in terms of sample efficiency. We find that SOLD consistently outperforms the highly sample-efficient DreamerV3 and TD-MPC2 baselines on all but the easiest *Reach-Specific* task, even after accounting for the samples used during pre-training. While the non-object-centric baseline demonstrates some success on the *Specific* tasks, it struggles with the relational reasoning required to solve the *Distinct* variants. In contrast, SOLD excels in tasks that demand reasoning about relationships between objects, as evidenced by the substantial performance gap observed on the *Distinct* tasks.

These results support our hypothesis that a structured latent representation within the world model significantly benefits tasks requiring object reasoning. This is especially valuable in robotics, where understanding object interactions is essential for solving complex control problems.

**Discovering Task-relevant Objects** To demonstrate that SOLD's improved relational reasoning capabilities are accompanied by an interpretable focus on task-relevant objects, we visualize an excerpt of the slot history in Figure 5. To illustrate the actor's attention pattern in the current (rightmost) time step, we multiply the attention scores by the masks of the respective objects and show them overlaid with the RGB reconstructions and as an individual colormap in the second and third rows, respectively. This visualization shows the Push-Specific task, where we find that the model automatically identifies task-relevant objects, disregarding slots that represent distractor objects across all time steps

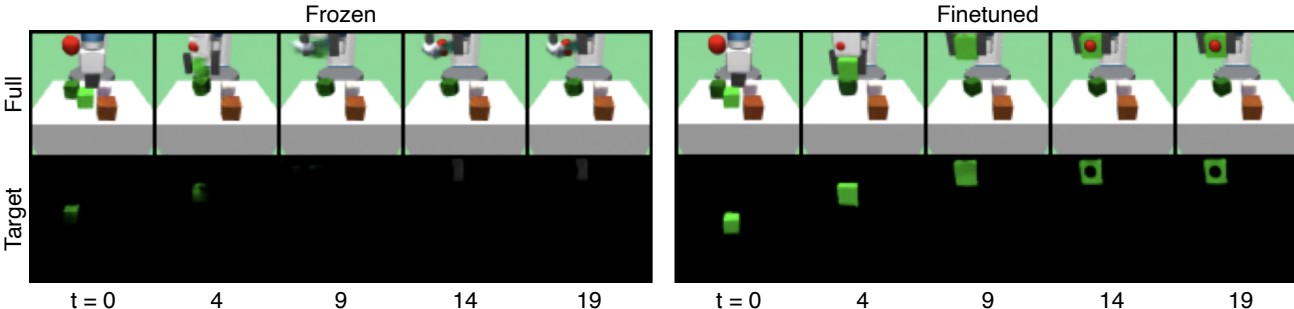

Figure 6: Decoded reconstructions of the full scene (top) and the slot corresponding to the target object (bottom) for a frozen (left) and finetuned (right) SAVi model. Finetuning enables the model to represent object configurations not present in the random pre-training data.

while focusing primarily on the robot and green cube. Although the recency bias induced by ALiBi is evident, we find that the model learns to overcome it when necessary, attending to the red sphere (indicating the goal position) after it has been occluded for 15 time steps, the last time it was visible. We see that the model effectively prioritizes task-relevant information, even when reasoning over long time sequences is required.

**SAVi Finetuning** A common limitation of prior work in object-centric RL is the reliance on encoder-decoder models pretrained on sequences with random behaviors and kept frozen during training, restricting their applicability to tasks where random and successful policies encounter similar state distributions. In the *Pick* tasks, this assumption is explicitly violated, as random behaviors rarely result in blocks being lifted off the table. Consequently, SAVi lacks prior exposure to configurations with blocks in the air, which will inevitably be reached by successful policies. Figure 6 illustrates the necessity of continually fine-tuning the object-centric encoder-decoder model. While the fine-tuned model accurately reconstructs lifted blocks, the frozen variant fails to decode this configuration correctly, causing the target block to dissolve during lifting. Additional examples are shown in Figure 15 in the Appendix.

### 4.3. Generalization to Non-Object-Centric Tasks

While object-centric methods are commonly applied to environments and datasets that naturally lend themselves to such decompositions, we aim to showcase the potential of our method to generalize beyond this setting. To this end, we train SOLD on the *Button-Press* and *Hammer* tasks from the Meta-World benchmark (Yu et al., 2019), both of which feature object with complex shapes and textures, challenging the model's ability to handle more diverse and visually intricate inputs. Additionally, we test SOLD on the *Cartpole-Balance* and *Finger-Spin* environments from the DM-Control suite (Tassa et al., 2018), which represent

significantly different domains not typically associated with object-centric learning. These environments are shown in Figure 7. SOLD achieves a 100% success rate on both the *Button-Press* and *Hammer* tasks, highlighting its ability to adapt to visually diverse and challenging object interactions. On the *Cartpole-Balance* and *Finger-Spin* tasks, SOLD achieves returns of 497 and 645, respectively, demonstrating its capacity to generalize to tasks where object-centric reasoning is less pronounced. Details about environment decompositions and dynamics predictions for all four tasks can be found in Section E.2 of the Appendix.

## 5. Related Work

**Object-Centric Learning** In recent years, the field of unsupervised object-centric representation learning from images and videos has gained significant attention (Yuan et al., 2023). Most existing methods follow an encoder-decoder framework with a structured bottleneck composed of $N$ latent vectors called slots, where each of these slots binds to a different object in the input image. Slot-based methods have been widely applied for images (Burgess et al., 2019; Locatello et al., 2020; Singh et al., 2021; 2023; Biza et al., 2023) and videos (Kipf et al., 2022; Singh et al., 2022; Elsayed et al., 2022; Bao et al., 2022). However, despite their impressive performance on synthetic datasets, they often fail to generalize to visually complex scenes. To overcome this limitation, recent methods propose using weak supervision (Elsayed et al., 2022; Bao et al., 2023), levering large pretrained encoders (Seitzer et al., 2023; Aydemir et al., 2023; Kakogeorgiou et al., 2024), or using diffusion models as slot decoders (Jiang et al., 2023).

**Object-Centric Video Prediction** Object-centric video prediction aims to understand the object dynamics in a video sequence with the goal of anticipating how these objects will move and interact with each other in future time steps. With this end, multiple methods propose to model and forecast the object dynamics using different architectures, in-

cluding RNNs (Zoran et al., 2021; Nakano et al., 2023) transformers (Wu et al., 2023; Villar-Corrales et al., 2023; Song et al., 2023; Daniel & Tamar, 2024; Nguyen et al., 2024) or state-space models (Jiang et al., 2024), achieving an impressive prediction performance on synthetic video datasets and learning representations that can help solve downstream tasks that require reasoning about objects properties and relationships (Wu et al., 2023; Petri et al., 2024). Biza et al. (2022) address the integration of action information into object-centric world models, but focus solely on action-conditional prediction and do not consider behavior learning through RL.

**Model-based RL**   Model-based approaches hold the potential to improve the sample efficiency of RL methods by learning environment dynamics, and recent years have seen several key contributions advancing this area. Pioneering work by Ha & Schmidhuber (2018) introduced the concept of a recurrent generative model, termed a world model, which captures the dynamics of visual RL environments. By encoding high-dimensional observations into a compact latent representation, this model enables RL agents to train policies entirely within imagined rollouts. The Planning Network (PlaNet) (Hafner et al., 2019) introduced a recurrent state-space model (RSSM) that predicts future states directly in a compact latent space, avoiding the costly step of decoding observations. PlaNet enables efficient planning of action sequences but is limited by the planning horizon. Building on this, Dreamer (Hafner et al., 2020) integrates planning and learning by training agents within a learned world model. Subsequent versions (Hafner et al., 2021; 2025) improved robustness and generalization through enhanced representation learning and optimization techniques. DreamerV3 has shown superior performance in visual control tasks compared to model-free approaches, but uses holistic rather than object-centric state representations. TD-MPC (Hansen et al., 2022) introduced a task-oriented latent dynamics model to optimize trajectories directly within the latent space of a world model. Unlike earlier approaches, TD-MPC avoids reconstructing observations, instead focusing the world model on reward-predictive elements through a loss applied to reward and value predictions. TD-MPC2 (Hansen et al., 2024) extends this work by introducing scalability improvements, demonstrating generalization abilities across multiple tasks and action spaces. Micheli et al. (2023) proposed IRIS, a method combining a discrete autoencoder with an autoregressive Transformer to model environment dynamics, demonstrating visually and temporally accurate predictions of game dynamics in Atari environments. While IRIS shares similarities with our approach – encoding an image into a set-based representation and predicting it forward using a Transformer – it lacks the object-centric interpretability afforded by our model.

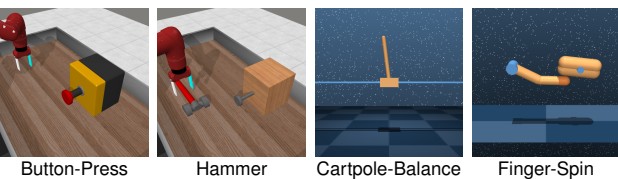

Button-Press    Hammer    Cartpole-Balance    Finger-Spin

Figure 7: Non-object centric environments from Meta-World (Yu et al., 2019) and DM-Control (Tassa et al., 2018).

**RL with Object-Centric Representations**   Recent works have explored integrating object-centric representations into RL frameworks. SMORL (Zadaianchuk et al., 2021) and EIT (Haramati et al., 2024) combined object-centric representations with goal-conditioned model-free RL for robotic manipulation. Yoon et al. (2023) investigated pre-training object-centric representations for RL, showing benefits for relational reasoning tasks. The field of object-centric model-based RL is still largely underexplored. Ferraro et al. (2023) introduce FOCUS, which learn object-centric world models for robotic manipulaiton. However, unlike our method, FOCUS does not use the object-centric states in forward prediction or action selection, but mainly to form an exploration target. Further, FOCUS requires supervision via ground-truth segmentation masks to learn the object-centric states. Ugadiarov et al. (2025) introduce ROCA, which learns an object-centric transition model from visual observations. However, unlike our method, this model us used to improve return estimates of the critic instead of learning through imagination inside world model rollouts.

## 6. Conclusion

We present SOLD, an object-centric model-based RL algorithm that learns directly from pixel inputs. By employing structured latent representations through slot-based dynamics models, our method offers a compelling alternative to traditional, holistic approaches. While object-centric representations have been valued for their role in forward prediction (Villar-Corrales et al., 2023), we demonstrate their synergistic benefits in accelerating the learning of behavior models. SOLD achieves strong performance across the introduced visual robotics environments, outperforming the state-of-the-art methods DreamerV3 and TD-MPC2, particularly in tasks requiring relational reasoning. Additionally, the learned behavior models exhibit interpretable attention patterns, explicitly focusing on task-relevant parts of the visual scene. To enable broader adoption, it will be interesting for future work to scale object-centric dynamics models to real-world data and highly stochastic environments. Ultimately, SOLD suggests that object-centric approaches hold strong potential for advancing model-based reinforcement learning.

## Impact Statement

This paper presents work whose goal is to advance the field of machine learning by introducing a novel object-centric model-based reinforcement learning algorithm, SOLD, designed for improved interpretability and sample efficiency. Our work has potential applications in robotics and automation, particularly in tasks requiring relational reasoning and manipulation. While we do not anticipate specific societal concerns arising directly from this work, the deployment of reinforcement learning systems in real-world applications warrants thoughtful consideration of ethical and safety challenges, including robustness and accountability in decision-making. Improved model interpretability, as explored in this work, could provide an avenue toward addressing these challenges.

## Acknowledgment

This work has partially been funded by grant BE 2556/16-2 (Research Unit FOR 2535 Anticipating Human Behavior) of the German Research Foundation (DFG) and by the German Federal Ministry of Education and Research (BMBF) in the project "Robotics Institute Germany (RIG)" under grant No. 16ME0999. Computational resources were provided by the German AI Service Center WestAI.

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

## A. Notation

### Slot Attention for Video (SAVi)

| | |
|---|---|
| $D_Z$ | The slot dimension |
| $N$ | The number of slots |
| $\boldsymbol{Z}_t$ | A set of slots $\boldsymbol{Z}_t \in \mathbb{R}^{N \times D_Z}$ at time-step $t$ |
| $\boldsymbol{Z}_{0:t}$ | A history of slot-sets up to time-step $t$ |
| $e_\eta$ | A SAVi encoder that maps $\boldsymbol{o}_t$ to $\boldsymbol{Z}_t$ |
| $d_\eta$ | A SAVi decoder that reconstructs $\boldsymbol{o}_t$ from $\boldsymbol{Z}_t$ |
| $\boldsymbol{F}_t$ | Features obtained by encoding images |
| $L$ | Number of spatial locations in $\boldsymbol{F}$ |

### Reinforcement Learning

| | |
|---|---|
| $S$ | The number of seed frames |
| $T$ | The imagination horizon |
| $\boldsymbol{o}_t$ | An image observation |
| $\boldsymbol{a}_t$ | An action command |
| $r_t$ | A reward |
| $R_t$ | A return |
| $\gamma$ | A scalar discount factor |
| H | The entropy of a probability distribution |
| $f_\alpha^{\mathrm{MLP}}$ | An MLP head that belongs to parameter group $\alpha$ |
| $\boldsymbol{h}_t$ | A processed output token |

# B. Implementation Details

Table 2: Implementation details for each of SOLD modules.

| (a) SAVi | | (b) Object-centric dynamics | | (c) Slot Aggregation Transformer | |
|---|---|---|---|---|---|
| **Hyper-Param.** | **Value** | **Hyper-Param.** | **Value** | **Hyper-Param.** | **Value** |
| Slot Dim. $D_Z$ | 128 | # Layers | 4 | # Layers | 4 |
| # Slots $N$ | 2-10 | # Heads | 8 | # Heads | 8 |
| Slot Init. | Learned | Token Dim. | 256 | Token Dim. | 256 |
| # Iters. | 3/1 | MLP Dim. | 512 | MLP Dim. | 512 |

In this section, we describe the network architecture and training details for each of the SOLD components. Our models are implemented in PyTorch (Paszke et al., 2019), have 12 million learnable parameters, and are trained on a single NVIDIA A-100 GPU with 40GB of VRAM. A summary of the model implementation details is listed in Table 2.

## B.1. Slot Attention for Video

We closely follow Kipf et al. (2022) for the implementation of the Slot Attention for Video (SAVi) decomposition model, including their proposed CNN-based encoder $e_\psi$ and decoder $d_\psi$, the transformer-based predictor and the Slot Attention corrector. We employ between 2 and 10 (depending on the environment) 128-dimensional object slots, whose initialization is learned via backpropagation. We empirically verified that learning the initial slots performs more stable than the usual random initialization. Furthermore, we use three Slot Attention iterations for the first video frame in order to obtain a good initial decomposition, and a single iteration for subsequent frames, which is enough to update the slot state given the observed features.

## B.2. Object-centric dynamics model

Our object-centric dynamics model is based on the OCVP-Seq (Villar-Corrales et al., 2023) architecture, which is a transformer encoder employing sequential and relational attention mechanisms in order to decouple the processing of temporal dynamics and interactions, and has been shown to achieve interpretable and temporally consistent predictions. We use 4 transformer layers employing 256-dimensional tokens, 8 attention heads, and using a hidden dimension of 512 in the feed-forward layers.

## B.3. Slot Aggregation Transformer

The *Slot Aggregation Transformer* (SAT) forms the architectural backbone for the reward, value and action models. This module aggregates information from object slots across multiple time steps to produce output tokens that are subsequently fed to MLP heads in order to predict rewards, values, or actions. An overview of our SAT module is depicted in Figure 8.

SAT is a causal transformer encoder module that receives as input a history of object slots, as well as a learnable output token `[out]` for each time step, which is responsible for producing the final output for the corresponding time step. Additionally, we append to the SAT inputs a number of register tokens `[reg]` per time-step, which have been shown to aid with processing in attention-based models by offloading intermediate computations from the output tokens and helping the module focus on relevant slots (Darcet et al., 2024).

To encode the positional information into SAT, we employ *Attention with Linear Biases* (Press et al., 2022) (ALiBi), which introduces linear biases directly into the attention scores, effectively encoding token recency. This approach helps the model deal with sequences of varying length, as well as generalize to longer sequences than those seen during training, thus outperforming absolute positional encodings.

For our experiments, SAT is implemented with 4 transformer encoder layers with causal self-attention, RMS-Normalization layers (Zhang & Sennrich, 2019), 8 attention heads, a token dimension of 256, and a hidden dimensionality in the feed-forward layers of 512. We set the number of learnable register token per time step to 4. Furthermore, we enforce in our causal attention masks that tokens belonging to time step $t$ cannot directly interact with previous output and register tokens.

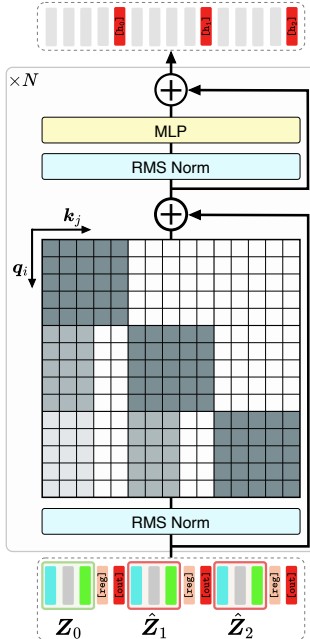

Figure 8: The *Slot Aggregation Transformer* applies causal masking and ensures that output and register token do not attend to themselves on other time-steps. The recency bias induced by ALiBi is visualized through the color gradient in the attention mask, with lighter shades of blue corresponding to a higher negative bias on the attention scores.

### B.4. Training Details

**SAVi Pretraining**  SAVi is pretrained for object-centric decomposition on approximately one million frames for 400,000 gradient steps. We use the Adam optimizer (Kingma & Ba, 2015), a batch size of 64 and a base learning rate of $10^{-4}$, which is first linearly warmed-up during the first 2,500 training steps, followed by cosine annealing (Loshchilov & Hutter, 2017) for the remaining of the training procedure. We perform gradient clipping with a maximum norm of 0.05.

**SOLD Training**  SOLD is trained using the Adam optimizer (Kingma & Ba, 2015) and different learning rates for each component: $10^{-4}$ for the dynamics and rewards models, and $3 \cdot 10^{-5}$ for training the action and value models, as well as for fine-tuning the SAVi encoder. To stabilize training, we perform gradient clipping with maximum norm of 0.05 for the SAVi model, 3.0 for the transition model, and 10.0 for the reward, value, and action models. For all components, we also use learning rate warmup for the first 2,500 gradient steps. Additionally, we implement the exponential moving average (EMA) for the target value network with a decay rate of 0.98. We use an imagination horizon of 15 steps for behavior learning, and the $\lambda$-parameter is set to 0.95.

# C. Baselines

In our experiments we compare our approach with three different baseline models, namely the state-of-the-art model-based RL algorithms *DreamerV3* and *TD-MPC2* and a *Non-Object-Centric* variant of our proposed model.

**DreamerV3**  DreamerV3 (Hafner et al., 2025) is a SoTA model-based reinforcement learning algorithm that learns behaviors from visual inputs without requiring task-specific inductive biases or extensive environment interaction. It builds a world model that predicts future states and rewards, which is then used to simulate potential outcomes and guide the agent's decision. DreamerV3 leverages latent dynamics and a compact, holistic representation of the environment for an efficient exploration, while showing desirable properties such as sample efficiency, scalability, and generalization across a wide range of complex tasks. We select the 12-million-parameter variant to match the parameter count of our proposed model. For further details, we refer to (Hafner et al., 2025).

**TD-MPC2**  TD-MPC2 (Hansen et al., 2024) builds upon its predecessor, TD-MPC (Hansen et al., 2022), with a series of architectural enhancements that improve scalability and sample efficiency. Like its predecessor, it avoids reconstructing high-dimensional inputs and instead focuses on modeling task-relevant dynamics in the latent space. The method employs temporal difference (TD) learning to predict future returns in the latent space and uses model predictive control (MPC) to optimize action sequences. Key advancements in TD-MPC2 include enhancements to training stability for larger model architectures and better generalization across tasks. These innovations allow it to achieve state-of-the-art performance on challenging visual and continuous control problems. For our experiments, we utilize the default 5-million-parameter variant since it is recommended by the authors for single-task RL problems.

**Non-Object-Centric Baseline**  This baseline model follows the same general framework as our proposed model, but replaces the object-centric SAVi encoder and decoder with a simple convolutional auto-encoder while keeping the remaining modules unchanged; thus allowing us to ablate the effect of object-centric representations for model-based reinforcement learning. The CNN auto-encoder used in this baseline consists of an encoder comprised of four strided convolutional layers with 64, 128, 256, and 512 channels respectively, each followed by batch normalization and a ReLU. The output of the final convolutional layer is flattened and fed through a linear layer to produce a 512-dimensional latent vector. The decoder mirrors the encoder structure, reconstructing the observations from the latent representation through the use of four transposed convolutional layers. To compensate for the lack of multiple latent vectors and to ensure a fair comparison, we increase the capacity of this baseline model by scaling the actor, critic, and dynamics models. Specifically, we increase the token dimension from 256 to 512, as well as the MLP hidden dimension from 512 to 1024. The total parameter count for this baseline is approximately 60 million, thus being five times larger than our proposed method and the DreamerV3 baseline.

# D. Environments

In this section, we provide further details about our proposed suite of environments, which includes eight object-centric robotic control tasks designed to test relational reasoning and manipulation capabilities. The environments, which are inspired by (Li et al., 2020) and are simulated using MuJoCo (Todorov et al., 2012), follow the same basic structure, consisting of a robot arm mounted on a base, positioned near a table where the manipulation tasks take place.

In all environments, the robot is controlled by a 4-dimensional action vector $\boldsymbol{a} = [a_x, a_y, a_z, a_{grip}] \in [-1, 1]^4$, where the first three components represent the desired movement direction of the end-effector, whereas the fourth component controls the opening and closing of the gripper. On the *Reach* and *Push* tasks, commands to the gripper are ignored, with the gripper fixed in a closed configuration, as gripping is not required to solve these tasks.

For all tasks, we define the following constants:

- $t_1 = 20$ and $t_2 = 10$: Temperature parameters that determine the steepness of the reward function.

- $d_m = 0.05$: Distance threshold (in meters) for considering a task successful.

**Reach**    In *Reach* tasks, the agent must identify a spherical target among several distractors and move the end-effector to its location. The reward is calculated as:

$$r = \exp(-t_1 \cdot ||\boldsymbol{p}_e - \boldsymbol{p}_t||_2), \tag{12}$$

where $\boldsymbol{p}_e$ is the position of the end-effector and $\boldsymbol{p}_t$ is the target position. Success is defined through the following condition at the last time step of an episode:

$$\text{success} = \begin{cases} 1 & \text{if } ||\boldsymbol{p}_e - \boldsymbol{p}_t||_2 < d_m \\ 0 & \text{otherwise} \end{cases}. \tag{13}$$

**Push & Pick**    Both *Push* and *Pick* correspond to reasoning and manipulation tasks where the agent must identify a single block among several distractors and move it to a target location. In *Push* tasks, the agent can slide the block to the target position on the table without using its gripper; whereas in *Pick* the target location can be above the table, thus requiring the agent to grasp the block in order to lift it to the target position. In both task variants the reward is calculated as:

$$r = 0.9 \cdot \exp(-t_1 \cdot ||\boldsymbol{p}_c - \boldsymbol{p}_t||_2) + 0.1 \cdot \exp(-t_2 \cdot ||\boldsymbol{p}_e - \boldsymbol{p}_c||_2), \tag{14}$$

where $\boldsymbol{p}_e$ is the position of the end-effector, $\boldsymbol{p}_t$ is the target position, and $\boldsymbol{p}_c$ is the block position. Success is defined through the following criterion, evaluated at the last time step of the episode:

$$\text{success} = \begin{cases} 1 & \text{if } ||\boldsymbol{p}_c - \boldsymbol{p}_t||_2 < d_m \\ 0 & \text{otherwise} \end{cases}. \tag{15}$$

# E. Additional Results

### E.1. Object-centric Decomposition

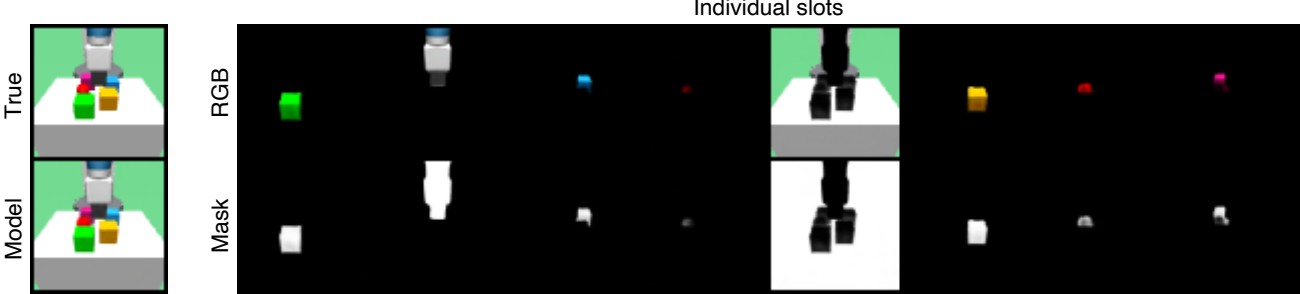

Figure 9: Object-centric SAVi decomposition of a video frame. We show the masked RGB image and the segmentation mask corresponding to each object slot. The masked RGB images are combined to reconstruct the observed frame.

Figure 9 depicts the object-centric decomposition of a video frame obtained by SAVi. SAVi parses the input frame into per-object RGB reconstructions and alpha masks, which can be combined via a weighted sum in order to accurately reconstruct the observed video frame. Notably, SAVi assigns an object slot to the scene background, five slots to different blocks, one slot to the red target, and one slot to the robot arm. The sharp object masks demonstrate that SAVi isolates object-specific information in each slot, which is beneficial for downstream applications such as behavior learning, allowing the agent to reason about object properties and their relationships while abstracting task-irrelevant details. Moreover, we find that SAVi is able to adapt to the varying number of objects in our environments, leaving extra slots empty when they are not needed to represent a scene.

### E.2. Open-loop Prediction

We visualize action-conditional open-loop predictions in the Push-Specific (Figure 10), Button-Press (Figure 11), Hammer (Figure 12), Cartpole-Balance (Figure 13), and Finger-Spin (Figure 14) environments.

Specifically, we present the ground truth sequence, predicted video frames, instance segmentation – where each object mask is assigned a distinct color – and object reconstructions for each slot.

In all examples, our model parses the scene into sharp, accurate object representations and models action-conditional object dynamics and interactions, enabling precise future frame predictions while maintaining object-centric representations.

We highlight SOLD's ability to capture complex physical interactions, such as pushing a block to a target location (Figure 10), pressing a button (Figure 11), or hammering a nail (Figure 12).

Furthermore, we demonstrate that SOLD generalizes to diverse, non-object-centric environments (Figure 13 and Figure 14), where sharp object separations and meaningful groupings emerge automatically – for instance, an object's shadow, despite being spatially distinct, is assigned to the same slot as the object itself.

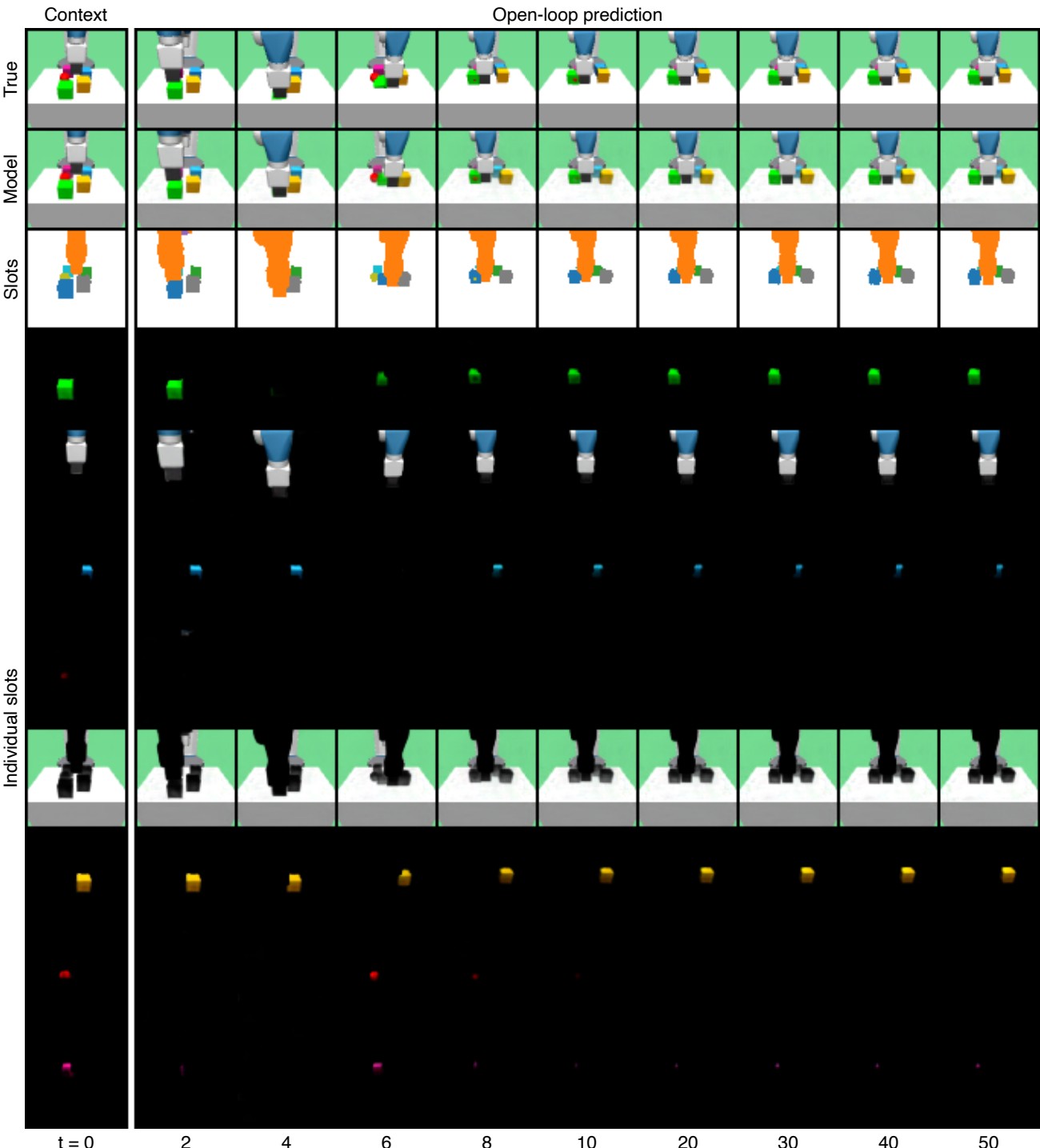

Figure 10: Open-loop prediction on the *Push-Specific* task. We visualize the ground truth, predicted frames, segmentation obtained by assigning different colors to each object mask, and per-object reconstructions. In this sequence, SOLD assigns one slot to the background, one slot for the robot, one slot for the target, and four different slots for blocks, while one slot remains empty.

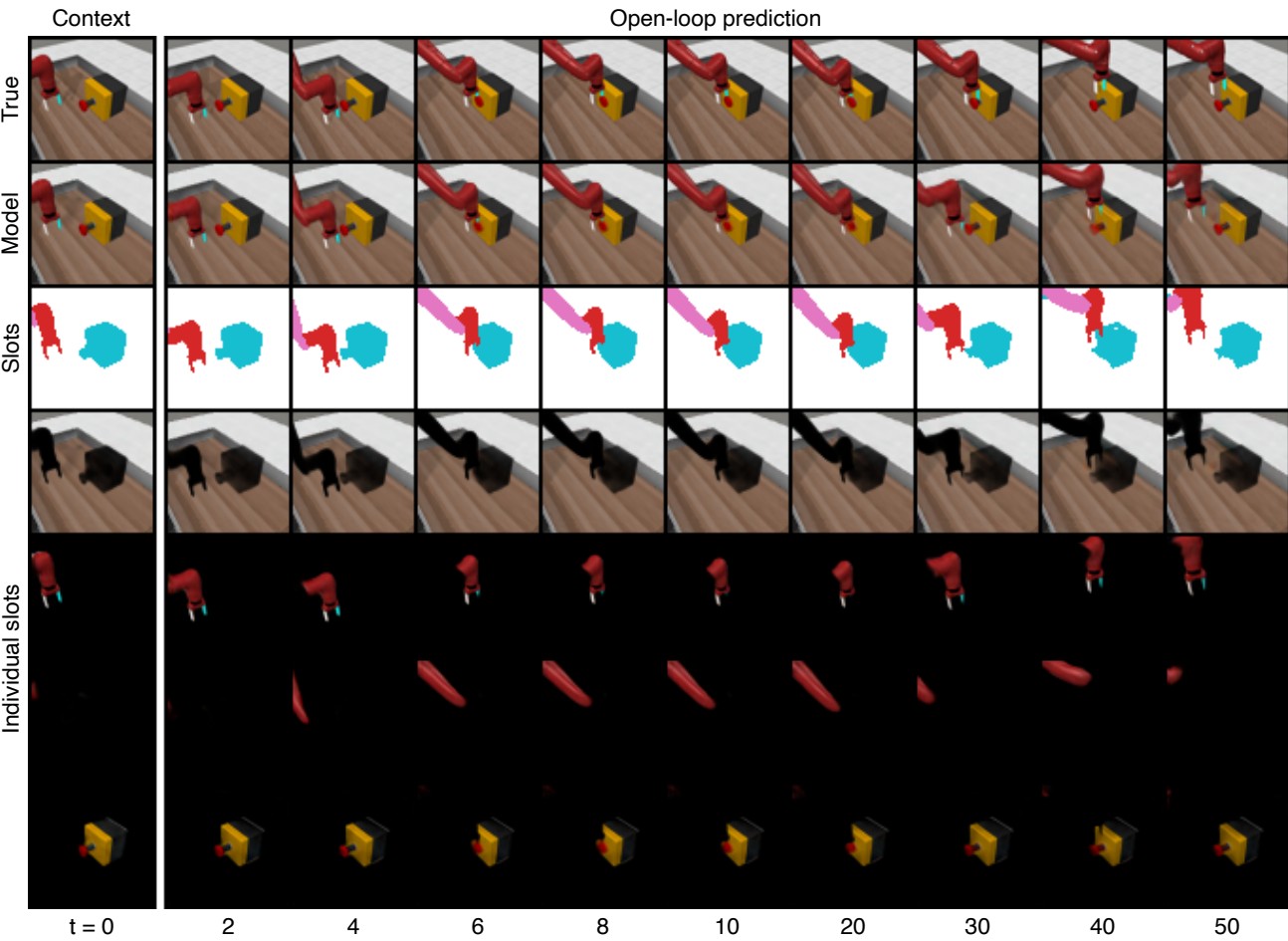

Figure 11: Open-loop prediction and decomposition results on the *Button-Press* task. We visualize the ground truth and predicted video frames, instance segmentation obtained by assigning a different color to each object mask, and per-object reconstructions. SOLD assigns a slot for the scene background, two slots for different robot parts, and a slot for the button-box.

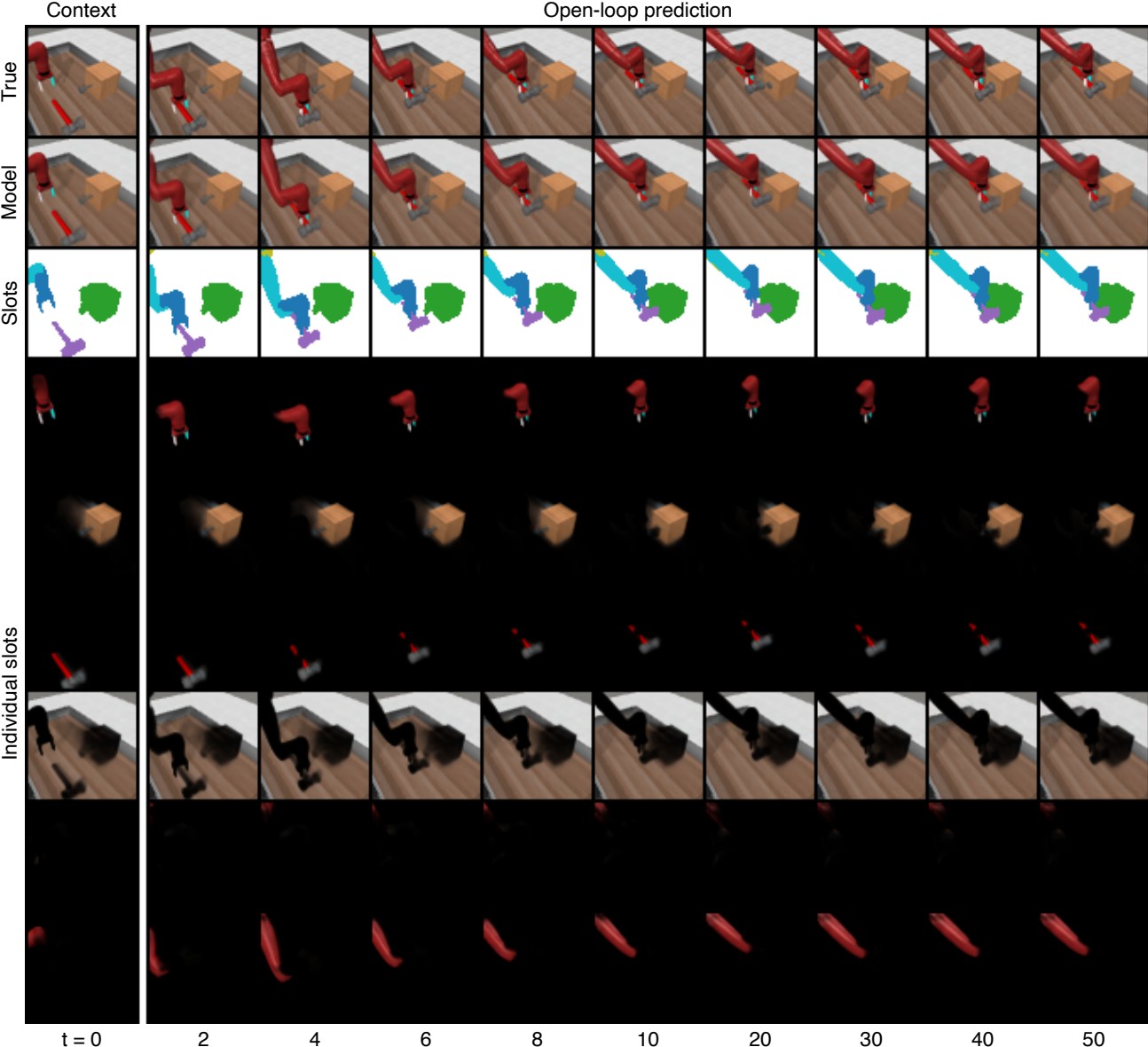

Figure 12: Open-loop prediction and decomposition results on the *Hammer* task. We visualize the ground truth and predicted video frames, instance segmentation obtained by assigning a different color to each object mask, and per-object reconstructions. SOLD assigns a slot for the scene background, three slots for different robot parts, a slot for the hammer, and a slot for the nail-box.

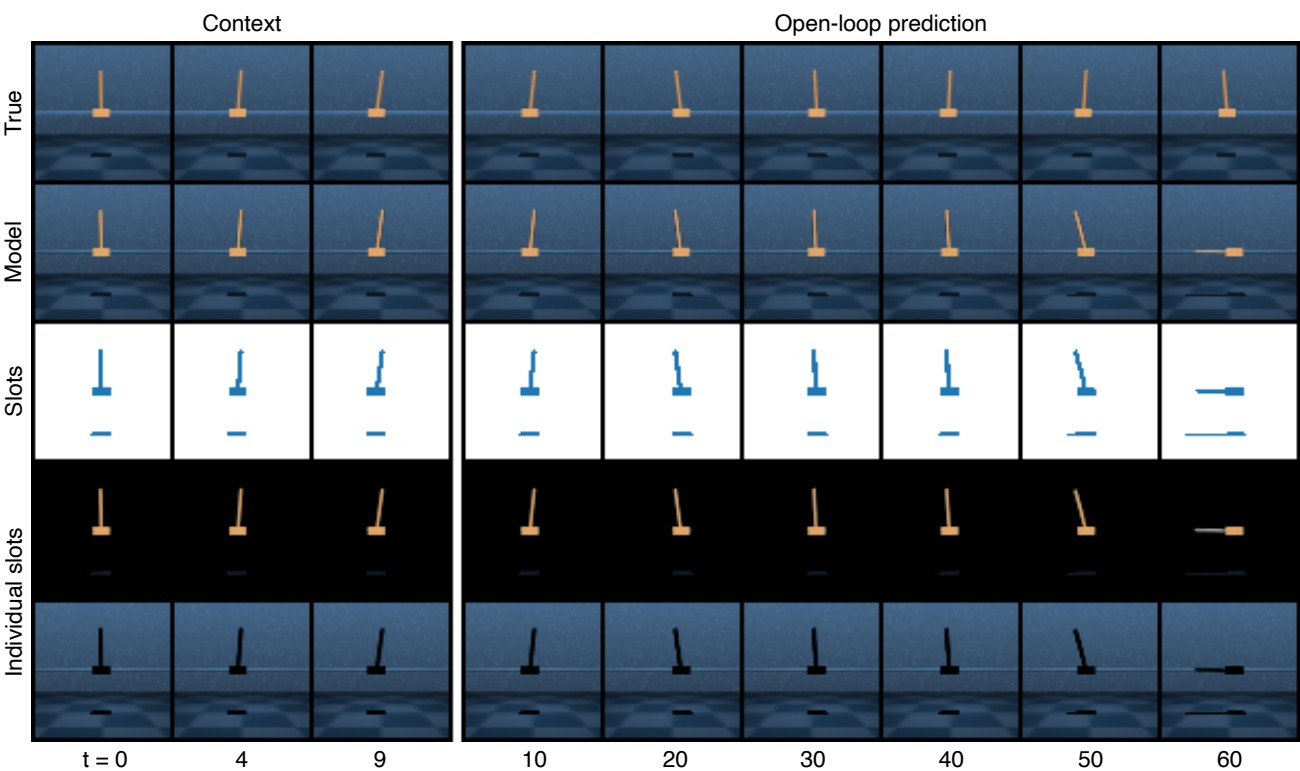

Figure 13: Open-loop prediction and decomposition results on the *Cartpole-Balance* task. We visualize the ground truth and predicted video frames, instance segmentation obtained by assigning a different color to each object mask, and per-object reconstructions. SOLD assigns a slot for the scene background and a slot for the cart-pole. Notably, the slot represents the object along with its shadow.

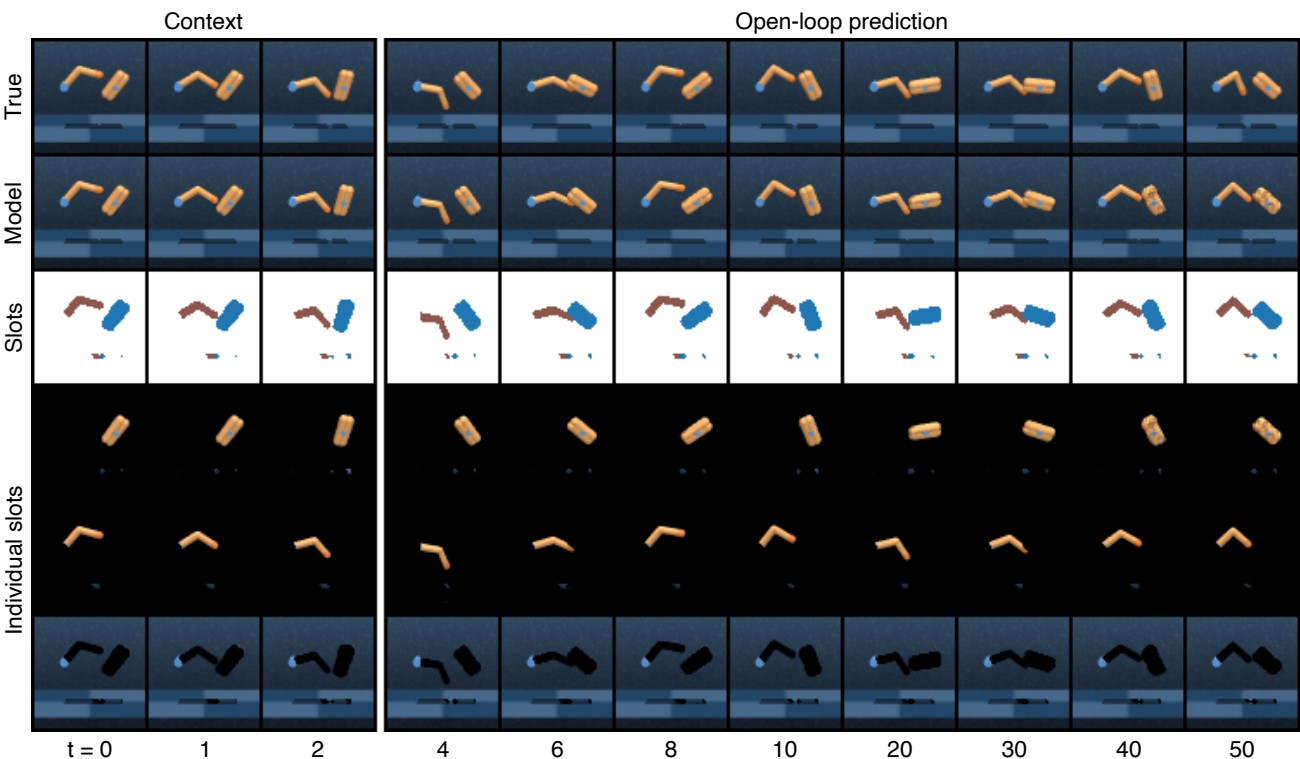

Figure 14: Open-loop prediction and decomposition results on the *Finger-Spin* task. We visualize the ground truth and predicted video frames, instance segmentation obtained by assigning a different color to each object mask, and per-object reconstructions. SOLD assigns a slot for the scene background, a slot for the finger, and a slot for the spinning target. Notably, the slots represent the objects along with their corresponding shadows.

### E.3. SAVi Fine-tuning

The *Pick* tasks highlight the need to design object-centric encoder-decoder modules that can adapt to changing state distribututions. Figure 6 exemplifies this challenge in the *Specific* variant, where the green target cube dissolves when lifted in the non-fine-tuned model.

Figure 15 underscores the importance of fine-tuning the object-centric encoder-decoder model with another example. Without fine-tuning, the blue color, which appears similarly on both colored blocks and the robot arm, leads to an even more drastic degradation of the reconstructions, where the robot itself is no longer accurately captured. In contrast, the fine-tuned model is able to reconstruct the sequence accurately.

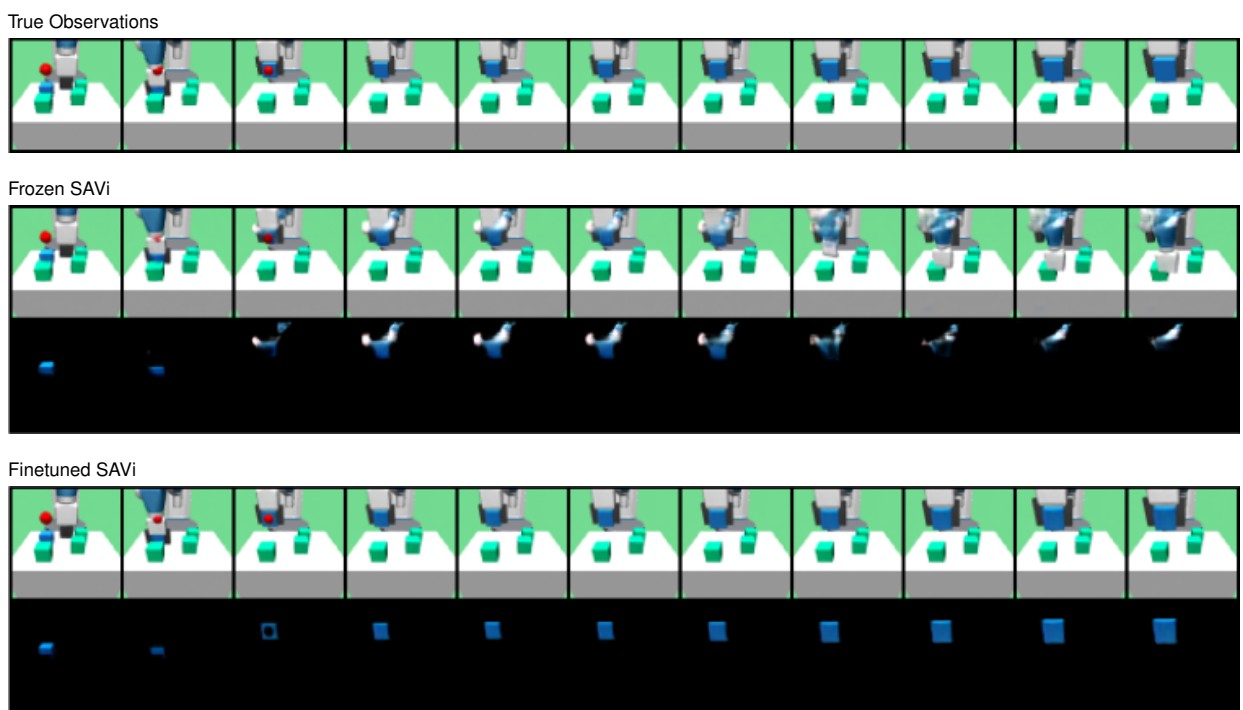

Figure 15: Comparison of fine-tuned and frozen SAVi models on *Pick-Distinct*. We visualize the full reconstruction and the slot that reconstructs the cube that is being lifted for both models. When the blue block is lifted off the table, the frozen model merges it with blue elements from the robot arm, deteriorating the prediction and hallucinating the arm going between the gripper fingers. The fine-tuned model, on the other hand, is able to reconstruct the sequence accurately.

## E.4. Discovering Task-relevant Objects

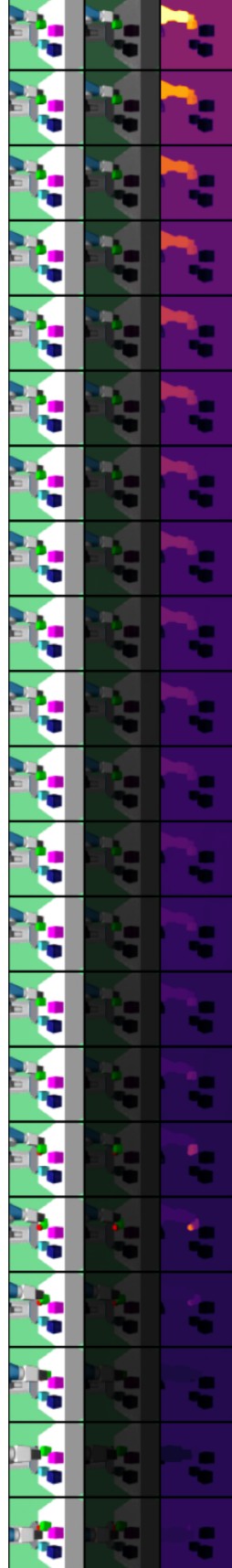

Figure 16: Original, uncut slot-history used in Figure 5. The full rollout highlights both the recency bias introduced by ALiBi and the model's ability to overcome this bias when task-relevant information must be retrieved from slots far in the past.

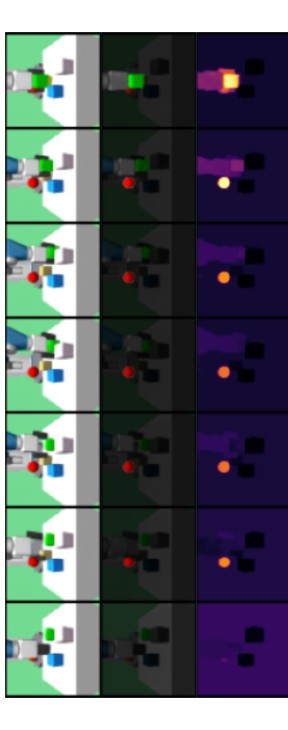

Figure 17: Visualizing the actor's attention over the slot-history on the *Pick-Specific* task reveals that the robot's gripper and target block in the current time-step receive the most attention, highlighting the model's ability to focus on task-relevant components. Further, the red target sphere is mostly occluded in the current time-step making it difficult to reconstruct its position accurately. However, the model learns to integrate information from previous steps, where it was still clearly visible.

