# OpenReview forum: "SOLD: Slot Object-Centric Latent Dynamics Models for Relational Manipulation Learning from Pixels"
_ICML.cc/2025/Conference — ICML 2025 poster_

### Official Review · Reviewer_BRLw · 2025-02-18

**Overall Recommendation:** 3

**Summary:**

The paper proposes a model-based RL (Dreamer-like) algorithm that utilizes pre-trained (and then fine-tuned) slot-attention based object-centric representations as the underlying state representation, in contrast to the standard single-vector representation (holistic) typically employed when learning from pixels. The proposed approach outperforms or performs comparably to recent model-based RL approaches.

**Claims And Evidence:**

* The paper claims model-based RL can improve sample efficiency over model-free RL, yet does not compare with one. In addition, there are sample-efficiency analyses.
* There are no comparisons with model-free RL object-centric approaches. While the authors cited several works like SMORL (https://arxiv.org/abs/2011.14381) and Haramati et al. (https://arxiv.org/abs/2404.01220v1), I believe an empirical comparison should be done, especially if the authors wish to claim their method is more sample-efficient (I’ll note that both methods are goal-conditioned, but it seems like it should be possible to apply them on the environments in this paper, and applying SOLD in their environments as well). I also believe that the environments in both mentioned works are more challenging than the ones used in this work.
* Generalizability: I’m worried about the generalizability of this method to more pixel-wise challenging tasks where SAVi (DDLP - https://arxiv.org/abs/2306.05957), and slot-attention in general (DINOSAUR - https://arxiv.org/abs/2209.14860), do not work well and fail to decompose the scenes.
* The claim (line 326) “This part-whole segmentation highlights the ability of slots to meaningfully identify and represent separate parts of a larger object, such as the gripper jaws of the robot” is not well-positioned. Do the authors claim that the ability to segment objects like humans (e.g., whole gripper instead of joints) is better for control? Usually state-based representations in robotics include the positions of the joints, so one could claim that aggregating all the joints to one object might actually hurt the performance.

**Essential References Not Discussed:**

I would like to point-out a concurrent work on object-centric world models: OC-STORM (https://arxiv.org/abs/2501.16443). However, the methods are very different.

**Experimental Designs Or Analyses:**

My only issue, as mentioned above, is that I find the experimental benchmark not challenging and that the chosen benchmark does not shed much light with respect to sample-efficiency.

**Methods And Evaluation Criteria:**

* The method makes sense for the problem, as well as the datasets.
* However, I find the environments and tasks used in this work quite simple (also visually simple) and more challenging environments/tasks would help provide more evidence for the claims in this work.

**Other Comments Or Suggestions:**

None

**Other Strengths And Weaknesses:**

**Strengths**:
* Open-source code (!)
* Very detailed appendix and a nice project site.
* Paper reads-well and easy to follow.

**Weaknesses/Limitations**:
* Reading all the training details, it seems the method is very brittle, and requires a lot of tuning (e.g., for learning rates and clipping values for each component).
* Performance: I like the idea of the paper and I believe there is a lot of promise in unsupervised object-centric representations, especially for control tasks, and I was surprised that the performance is not much better compared to non-object-centric baselines in most environments. I do wonder if the problem is the choice of easy/simple tasks/environments or the choice of underlying object-centric representations (slot-attention).

**Questions For Authors:**

* (Repeated from above) The claim (line 326) “This part-whole segmentation highlights the ability of slots to meaningfully identify and represent separate parts of a larger object, such as the gripper jaws of the robot” is not well-positioned. Do the authors claim that the ability to segment objects like humans (e.g., whole gripper instead of joints) is better for control? Usually state-based representations in robotics include the positions of the joints, so one could claim that aggregating all the joints to one object might actually hurt the performance.
* Line 401: the authors report the returns on Cartpole-Balance and Finger-Spin, but without any comparison these numbers are meaningless. Should the reader have knowledge of other algorithms’ performance on these tasks?
* Appendix, line 737: what does “whose initialization is learned via backpropagation” mean? How does one learn initialization?
* Ablations: have the authors ablated the choice of positional encoding, or just used AliBi by default? Similarly, the number of register tokens?
* What is the effect of freezing/fine-tuning the SAVi backbone quantitatively?
* Stability: from personal experience, Slot-Attention can be very hard to get working (as also been shown in many previous works, like DINOSAUR and DDLP mentioned above), and not very consistent (i.e., different runs with same hyper-parameters but sometimes Slot-Attention does not provide a good decomposition to slots). This is more evident on more visually complex/real-world datasets. I’m curious regarding the authors’ experience with SAVi’s stability in that sense, was it common to have runs where SAVi did not provide good decompositions (e.g., multiple objects in a single slot)?
* How long does it take to train the model (roughly, wall-clock hours)?

**Relation To Broader Scientific Literature:**

* The main contribution of this work is extending slot-based video prediction models (OCVP) with actions and reward to make them world-models.
* While I find this a solid contribution, I don’t think the performance of the proposed method, nor the chosen benchmark, are impressive enough to be convinced regarding the claims and the promise of the method.

**Theoretical Claims:**

There are no theoretical claims in this paper.

---

> ### Author Rebuttal · Authors · 2025-04-01
>
> We want to thank the reviewer for appreciating the potential of our idea and the clarity of the paper. We address your questions below.
>
> ***"I’m worried about the generalizability of this method to more pixel-wise challenging tasks where SAVi fails to decompose the scenes."***
>
> Please see our response to Reviewer tPrN (Question 2) regarding challenges with OCRs in complex scenes and potential future work. SOLD itself is designed to be agnostic to the specific OC model, allowing it to benefit from future OCR improvements.
>
> ***"The claim (line 326) 'This part-whole segmentation highlights the ability of slots to meaningfully identify and represent separate parts of a larger object, such as the gripper jaws of the robot' is not well-positioned."***
>
> Our intention with the phrasing “part-whole segmentation” was to highlight SAVi's ability to learn nuanced representations by assigning distinct slots to the gripper jaws (visualized in brown/gray in Figure 3) separate from the main robot arm body (orange). We view this emergent separation as beneficial, particularly as the gripper jaws have independent actuation via a dedicated component of the action vector, making their distinct representation potentially valuable for learning control. We did not intend to claim this specific granularity is universally superior, and indeed, as shown in Figures 10 and 11 in the Appendix, SAVi can represent the robot arm at varying levels of subdivision depending on the visual prominence and context. We will revise the phrasing in the paper for better clarity.
>
> ***“I find the environments and tasks used in this work quite simple (also visually simple) and more challenging environments/tasks would help provide more evidence for the claims in this work.”***
>
> While we intentionally limit the visual complexity to study control learning when OCRs are obtainable with current methods (see also our response to Question 1 of Reviewer o1iz), we agree that addressing also more visually complex scenarios will be an interesting avenue for future work.
>
> **Why is the performance ***“not much better compared to non-OC baselines in most environments”***?**
>
> While SOLD outperforms baselines overall and especially on distinct tasks (Table 1), we deliberately included easier tasks (Reach/Push-Specific) to paint a fuller picture of the level of complexity at which OCRs yield payoffs and to be fair to the baseline methods. We found that the upfront cost of pre-training OCRs pays off as complexity increases. On the easiest Reach-Specific task, TD-MPC2 converges quickly (within the SAVi pre-training window), making the effort to acquire OCRs less warranted *for that specific task.* TD-MPC2 (which operates reconstruction-free) can rapidly extract the minimal necessary information (target/end-effector position) to solve the task.
> We feel that leaving out such configurations would distort the picture of where significant improvements due to OCRs, accounting for the cost of both pre-training and behavior learning, were made.
>
> ***“What does 'whose initialization is learned via backpropagation' mean?”***
>
> This refers to having a parameter vector to which the slot is initialized, which is trained with back-propagation. We link the implementation at https://anonymous.4open.science/r/sold-rebuttal/slot_initialization/learned.py
>
> ***“Ablations: have the authors ablated the choice of positional encoding, or just used AliBi by default?”***
>
> We added ablations at https://anonymous.4open.science/r/sold-rebuttal/ABLATIONS.md
>
> ***“I’m curious regarding the authors’ experience with SAVi’s stability [...]?”***
>
> Please see our answer to Reviewer tPrN (Question 3).
>
> **How long does it take to train the model?**
>
> On a single A100 (40GB): SAVi pre-training takes ~2-3 days. SOLD training takes up to ~10 days for the longest tasks (Push/Pick, 50 steps per episode). For comparison, DreamerV3 (optimized JAX code, holistic repr.) took ~4.5 days.
>
> **What is the effect of freezing/fine-tuning the SAVi backbone quantitatively?**
>
> We added experiments comparing these two settings quantitatively:
> https://anonymous.4open.science/r/sold-rebuttal/SAVI_FINETUNING.md
> Interestingly, there is only a small drop in performance from the fine-tuned to the frozen SAVi model. We hypothesize this to be due to the SAT model, which can leverage a long history of slots, compensating for deteriorating representations.
> However, we chose to emphasize again the strong qualitative results in Figures 14 and 15 because it clearly indicates a potential failure mode of current OCRL approaches (e.g., SMORL, EIT), which pre-train the OC model and freeze it for the downstream RL task. We believe highlighting this issue and exploring fine-tuning as a strategy to maintain representation quality under distribution shift is an important discussion point for the OCRL community.
>
> ---
>
> We hope our responses have adequately addressed your concerns, and we would be grateful if you consider updating your score.

---

> > ### Comment · Reviewer_BRLw · 2025-04-03
> >
> > Thank you for your response and clarifications. After reading the response, the other reviewes and responses, some of my concerns regarding ablations have been addressed, but the concerns regading the choice of the object-centric algorithm and performance remain. In addition, it seems that there is a general agreement between reviewers regarding the weaknessess, and specifically, it seems that reviewer tPrN and I are aligned. As such, I'm going to keep my score for now.

---

> > > ### Author Response · Authors · 2025-04-07
> > >
> > > We are pleased to have addressed your concerns regarding the SAT architecture ablations and SAVi fine-tuning. We appreciate the opportunity to address the performance of SOLD compared to the baseline methods, as well as the choice of the OC algorithm below.
> > >
> > > **Regarding the choice of the OC encoder-decoder framework.**
> > >
> > > In the following we outline why we deliberately choose to use SAVi as the OC encoder-decoder method in our work.
> > > The goal of our research is to provide insight into the potential of MBRL from OCRs when such representations can be well learned with current methods. In addition to SAVi, we have experimented with the use of STEVE (https://arxiv.org/pdf/2205.14065) and VideoSAUR (https://arxiv.org/pdf/2306.04829). However, we found that the size of the pre-training dataset was the central factor in determining the quality of the decompositions for all three methods.
> > > Because STEVE and VideoSAUR add complexity to the OCR learning problem and make it difficult to verify the isolation of object-specific information in each slot (this is because STEVE uses an autoregressive transformer decoder instead of predicting per-slot alpha masks, and VideoSAUR predicts ViT embeddings at the patch level), we found SAVi to be the most appropriate model to achieve the goals of our investigation. Namely, to study behavior learning from OCRs that sufficiently disentangle visual scenes, and to explore interpretable patterns in the learned attention weights of the model. To this end, we achieved the goal of demonstrating improved relational reasoning and interpretability.
> > >
> > > In summary, while SOLD itself is designed to be agnostic to the specific OCR learning method used, beyond the ability to convert visual observations into a set of vectors, we explicitly considered this decision and are convinced that SAVi is the best method for the goals of this investigation. However, we agree that extending our work to real-world and visually complex data is an interesting avenue where the agnostic design of SOLD will be beneficial. Nevertheless, we consider this to be a distinct investigation of learning from complex, realistic visual data, separate from the goals of this work.
> > >
> > >
> > > **Regarding the performance compared to the baseline methods.**
> > >
> > > The overall performance gap between SOLD (82% success rate) and the second best method DreamerV3 (56.4% success rate) is just over 25%, which we consider a significant improvement. This performance gap is achieved despite our choice to include simpler problems, such as the Specific task variants, where limited OC reasoning is required to solve the control problem. Nevertheless, we are able to show that SOLD significantly outperforms SoTA non-OC methods across the considered task suite. Moreover, SOLD is able to learn the Reach-Distinct-Groups task, where no baseline method currently makes progress, which speaks to its difficulty and the ability of SOLD to open up new relational reasoning capabilities.
> > >
> > > While we agree that considering even more complex tasks is a valuable investigation for future work, we believe that this outperformance is significant and demonstrates the utility of SOLD’s OC structure.
> > >
> > > ---
> > >
> > > Finally, we want to thank the reviewer for their detailed feedback on our work, which has helped us to clarify the rationale for design and evaluation decisions regarding both the OC encoder-decoder model and the subsequent behavior learning. Since the reviewer mentions to be in agreement with reviewer tPrN regarding remaining weaknesses, we also kindly point to our reply to the rebuttal comment of reviewer tPrN, which discusses remaining limitations and open questions in detail.
> > >
> > > We sincerely hope that our response addresses your remaining concerns. If it does, we would greatly appreciate your consideration in updating your score.

---

### Official Review · Reviewer_tPrN · 2025-03-12

**Overall Recommendation:** 3

**Summary:**

The manuscript introduces the slot object-centric latent dynamics (SOLD) model, a reinforcement learning (RL) algorithm that leverages an object-centric latent world model, which is learned directly from pixels, for behavior learning. The world model is an extension of object-centric video prediction (OCVP), a model that leverages slot attention for video (SAVi) for object centric representation discovery and a transformer-based prediction model with two attention mechanisms for separately capturing temporal information of the same object and relational information about interactions between objects. The world model in SOLD differs from OCVP in that 1) it conditions the transformer’s predictions on past and current actions in addition to the observation history, 2) SAVi is pre-trained on random observation sequences from random episodes, and 3) a reward prediction model is added to enable behavior learning purely from imagined trajectories.
For behavior learning SOLD uses a very similar approach to DreamerV3. The proposed method is evaluated against DreamerV3, TD-MPC2, and SOLD w/ a CNN encoder instead of SAVi on a suite of eight object-centric robotic control environments, exhibiting superior performance. The method is also shown to work on environments that are not designed with object-centric learning in mind.

## update after rebuttal
The authors' rebuttal addressed some of my concerns. However, I share the concern with other reviewers that the evaluation is no thorough enough. For instance, the authors mentioned that hyperparameters for baselines were copied from prior work which used different environments, while for SOLD some hyperparameters are tuned. Comparison on more established benchmarks would make the results more convincing. I still believe the work contributes valuable insight for the community and lean towards acceptance.

**Claims And Evidence:**

* The model is shown to work well on the considered relational reasoning tasks.
* “We introduce SOLD, the first object-centric MBRL algorithm to learn directly from pixel inputs [...]”
I believe several works mentioned in the “Essential References Not Discussed” section of this review are implementing object centric world models that are trained with pixel supervision. Please clarify or adjust this claim.
* Regarding SOLD outperforming state-of-the-art methods (SOTA): Those methods are SOTA on other benchmarks than the ones used in this study. The claim should be more narrow (i.e. on a new benchmark suite focusing on relational reasoning) or supported by additional comparisons on widely-used benchmarks.

**Essential References Not Discussed:**

Given the state of object-centric representations being underexplored in RL, as mentioned in the paper, I’d suggest to briefly discuss the relation to the following works:
1. Biza, O., Platt, R., van de Meent, J. W., Wong, L. L., & Kipf, T. Binding Actions to Objects in World Models. In ICLR2022 Workshop on the Elements of Reasoning: Objects, Structure and Causality.
2. Collu, J., Majellaro, R., Plaat, A., & Moerland, T. M. (2024). Slot Structured World Models. arXiv preprint arXiv:2402.03326.
3. Heravi, N., Wahid, A., Lynch, C., Florence, P., Armstrong, T., Tompson, J., ... & Dwibedi, D. (2023, May). Visuomotor control in multi-object scenes using object-aware representations. In 2023 IEEE International Conference on Robotics and Automation (ICRA) (pp. 9515-9522). IEEE.
4. van Bergen, R. S., & Lanillos, P. (2022, September). Object-based active inference. In International Workshop on Active Inference (pp. 50-64). Cham: Springer Nature Switzerland.
5. Zadaianchuk, A., Seitzer, M., & Martius, G. (2020). Self-supervised visual reinforcement learning with object-centric representations. arXiv preprint arXiv:2011.14381.

**Experimental Designs Or Analyses:**

* I could not find a description of how hyperparameters were selected
* Appendix E mentions that the capacity of the Non-Object-Centric Baseline was increased for fair comparison. Did you evaluate whether that actually helps its performance? Model size or capacity is not always the best criterion for fairness. Is the CNN a standard architecture used in similar environments?

**Methods And Evaluation Criteria:**

At a high level SOLD replaces the DreamerV3 world model with one based on OCVP. The method is compared with DreamerV3, TD-MPC2 and SOLD w/o object-centric representation on a suite of robotic control tasks that require basic relational reasoning skills. The level of relational reasoning is limited to one-vs-all comparisons of color or spatial relation between the arm and one or two objects. How does SOLD compare with the baselines on the two tasks from the Meta-World benchmark or other more commonly used tasks?

**Other Comments Or Suggestions:**

The below citation only shows the first of multiple authors without “et al.”:

Cho, K. Learning phrase representations using RNN encoder-decoder for statistical machine translation. In Conference on Empirical Methods in Natural Language Processing (EMNLP), 2014.

**Other Strengths And Weaknesses:**

* The paper is very well organized and explains motivation, methodology and experimental setup clearly.
* Providing performance comparisons with the baselines on more established benchmarks and a discussion of more works in the intersection of object-centric representations and RL would significantly improve the paper and I'd probably increase the score.

**Questions For Authors:**

On potential limitations:
* How could one address the problem of slot attention potentially not separating objects very well in more complex environments? What other methods for object-centric representations have you considered?
* Does slot attention occasionally collapse to degenerate solutions?
* Have you considered partially observable environments (beyond simple occlusions)?

**Relation To Broader Scientific Literature:**

According to my understanding the present manuscript is not the first to propose slot-based world models. The method’s differences to other object-centric world models should be discussed in (more) detail.

**Theoretical Claims:**

N/A

---

> ### Author Rebuttal · Authors · 2025-04-01
>
> We want to thank the reviewer for acknowledging the clarity of our paper and methodology and for the valuable references, which we will discuss in the final version. In the following, we aim to address your questions.
>
> ***"The method’s differences to other OC world models should be discussed in (more) detail."***
>
> While we discuss these references in detail in the final version, we want to briefly point out differences to our method here:
> - While Bize et al. (https://arxiv.org/pdf/2204.13022) consider the problem of adding actions to an OC world model, they do so only for action-cond. prediction, not for behavior learning with MBRL (“A dataset of expert demonstrations is available [...] and the model is evaluated on its ability to predict the block positions”, Section 4.3).
> - Collu et al. (https://arxiv.org/pdf/2402.03326) learns a slot-attention (SA)-based OC dynamics model. While it is referred to as a world model, no reward prediction, action-conditioning or behavior learning is done.
> - Heravi et al. (https://arxiv.org/pdf/2205.06333) use OCR learning on robotic manipulation tasks, showing that SA-based representations yield performance improvements for object location prediction. However, they consider neither world models nor RL (“We only consider the imitation learning setup and learn a policy from a dataset of demonstrations”, Section IV.E).
> - van Bergen et al. (https://arxiv.org/pdf/2209.01258) consider the prediction of video frames on the basis of OCRs but follow the active inference framework.
> - While Zadaianchuk et al. (https://arxiv.org/pdf/2011.14381) use SCALOR to infer OCRs from pixel inputs, they use model-free RL to learn their policies (“our policy can be trained with any goal-conditional model-free RL algorithm”, Section 4.2).
>
> We want to make sure to depict the novelty of our method accurately and to give credit to all prior works, so we would greatly appreciate the reviewer’s opinion on the rephrased contribution to differentiate SOLD from prior works given in response to Reviewer kT4P (Question 1).
>
> ***"How could one address the problem of SA potentially not separating objects well in more complex environments? What other methods for OCRs have you considered?"***
>
> While the objective of our research is to provide insights into the potential of MBRL from OCRs when such representations can be learned well with current methods, rather than to improve the OCR learning method itself, we concur that learning meaningfully separated objects on complex visual data is a central point for expanding the range of applications of our method.
> In addition to SAVi, we have experimented with using STEVE (https://arxiv.org/pdf/2205.14065) and VideoSAUR (https://arxiv.org/pdf/2306.04829). However, we observed that the size of the pre-training dataset was the central factor in determining the quality of the decompositions for all three methods.
> Because STEVE and VideoSAUR add complexity to the OCR learning problem and make it challenging to verify the isolation of object-specific information in each slot (this is because STEVE uses an autoregressive transformer decoder instead of predicting per-slot alpha masks, and VideoSAUR predicts ViT embeddings at the patch level), we found SAVi to be the most suitable model to achieve the goals of our investigation. Namely, to study behavior learning from OCRs that sufficiently disentangle visual scenes, and to explore interpretable patterns in the model's learned attention weights.
> However, there are several avenues to extend our work in the future to apply it to visually complex data, including the prediction of ViT embeddings instead of pixels (especially for larger images), and the use of vision-foundation models to guide slot initializations.
>
> **Does SA occasionally collapse to degenerate solutions?**
>
> SAVi training can occasionally collapse (e.g., poor object separation despite good reconstruction) due to misconfiguration (not enough slots), insufficient pre-training data, or seed dependence. See examples: https://anonymous.4open.science/r/sold-rebuttal/SAVI_FAILURE_CASES.md.
> However, we found SAVi robust during fine-tuning once converged, which is valuable for MBRL.
>
> **Have you considered partially observable environments?**
>
> Occlusions have induced the demand for challenging long-horizon reasoning for the considered tasks. A key objective of our work was to ascertain whether the proposed models are successfully able to resolve the resulting ambiguities.
> Additionally, occlusion, as an object-dependent source of partial observability, enables interpretable visualization of the attention patterns to inspect how the model is retrieving missing information. Investigating other sources of partial observability is important future work, especially for real-robot deployment.
>
> ---
>
> We hope that we have been able to address open questions and better situate SOLD in the OC-RL landscape. If so, we would be thankful if you considered updating your score.

---

> > ### Comment · Reviewer_tPrN · 2025-04-02
> >
> > Thank you for providing these thoughtful clarifications! I'm still hesitant to increase the score, since the authors did not comment on the following items mentioned in my review:
> > 1. Regarding SOLD outperforming state-of-the-art methods (SOTA): Those methods are SOTA on other benchmarks than the ones used in this study. The claim should be more narrow (i.e. on a new benchmark suite focusing on relational reasoning) or supported by additional comparisons on widely-used benchmarks.
> > 2. Providing performance comparisons with the baselines on more established benchmarks
> > 3. I could not find a description of how hyperparameters were selected
> > 4. Appendix E mentions that the capacity of the Non-Object-Centric Baseline was increased for fair comparison. Did you evaluate whether that actually helps its performance? Model size or capacity is not always the best criterion for fairness. Is the CNN a standard architecture used in similar environments?

---

> > > ### Author Response · Authors · 2025-04-07
> > >
> > > We are glad the reviewer found our clarifications helpful. We were constrained by space and appreciate the opportunity to address the remaining questions.
> > >
> > > **The claim regarding outperformance of SoTA methods should be more narrow or supported by additional comparisons on widely used benchmarks.**
> > >
> > > We want to thank the reviewer for raising this important point, helping us to ensure that our claims reflect the obtained results accurately.
> > >
> > > We do not mean to claim that SOLD is universally superior to SoTA RL methods. Instead, akin to prior works investigating OCRs in RL (https://arxiv.org/abs/2302.04419, https://arxiv.org/abs/2011.14381, https://arxiv.org/pdf/2404.01220), SOLD is able to make progress on *specific but crucial* abilities, namely relational reasoning and interpretability.
> > >
> > > While interpretability is not easily quantifiable, we are encouraged that all reviewers highlighted and appreciated this aspect of our work - an aspect that is rarely considered in RL approaches and where we believe SOLD adds value to the current MBRL landscape.
> > >
> > > Relational reasoning, on the other hand, is a measurable skill and long recognized as a cornerstone of human intelligence. To evaluate this, we introduced the proposed benchmark. Importantly, for tasks that require reasoning over multiple objects, we are able to make improvements to the efficiency with which such tasks can be learned, which become more pronounced as the difficulty of the task increases.
> > >
> > > To accurately reflect these results, we will revise the description of our contribution according to the reviewer's suggestions to make these aspects more explicit in the final version. In addition, we will ensure that the role of the non-OC tasks studied is more clearly outlined as a means of demonstrating the potential of our OC dynamics model to generalize to such scenes, not that OC reasoning generally is superior for solving the associated control problems.
> > >
> > > **How were the hyperparameters (HPs) selected?**
> > >
> > > Since an exhaustive HP search is not feasible, we indicate values chosen by us through experimentation in square brackets, with bold numbers indicating the chosen values, while other HPs were adopted based on previous work, with the rationale explained following the respective tables.
> > >
> > > | SAVi HPs |Value|
> > > |:---|:---|
> > > |Slot Dim. $D_Z$| [64, **128**, 256]|
> > > |Num. Slots $N$|2-10|
> > > |Slot Initialization|[Learned-Random, **Learned**]|
> > > |SA Iterations|3/1 |
> > >
> > > We consider both SAVi and OCVP to select standard HPs. These include the number of SA iterations, the architecture of the models employed by SAVi, and the batch size used for training.
> > > We run experiments with both initialization strategies that SAVi introduces and find that Learned leads to better specialization of the slots and thus better decomposition of the visual scene, i.e., the isolation of information in the slots, a property we are interested in to study control learning and to interpret how information is retrieved by the model, was better on our problems.
> > >
> > > |Dynamics Model HPs|Value|
> > > |:---|:---|
> > > |Num. Layers|4|
> > > |Token Dim.|256|
> > > |Residual|True|
> > > Teacher-forcing|False|
> > >
> > > For the OC dynamics model, we select HPs on the basis of ablations already performed in the OCVP paper, which evaluated residual vs absolute prediction of slots, the use of teacher-forcing, and the number transformer layers (Table 3 in the OCVP paper), which we use as a reference for our act-cond. prediction model. Since we found this model to be capable of accurately modeling OC dynamics, no further HP search was performed on its architecture.
> > >
> > > For the SAT, we use the OC dynamics model as the reference for the standard HPs. Additionally, we have added ablations regarding the employed positional encoding and the prediction of scalar values.
> > >
> > > ***”[...] the capacity of the Non-OC Baseline was increased for fair comparison. Did you evaluate whether that actually helps its performance? Is the CNN a standard architecture used in similar environments?”***
> > >
> > > Yes, the baseline is a standard CNN autoencoder architecture similar to that used in SAVi or DreamerV3. While the performance difference to SOLD is substantial, we agree that it is helpful to ablate whether capacity is the best metric for a fair comparison and we are running ablations to compare the performance of using a larger vs standard SAT architecture. At this time, it is too early to evaluate the exact results, but the performance appears to be similar. For the final version, we will add whichever method performs better and explain this ablation in the Appendix.
> > >
> > > Finally, we want to thank the reviewer for their detailed feedback which has been substantially helpful to better situate SOLD in the full landscape of related work, ensure our claims precisely reflect the obtained results, and add experiments and details to clarify questions.
> > > We sincerely hope that our response has addressed your remaining concerns, and we would greatly appreciate your consideration in updating your score.

---

### Official Review · Reviewer_o1iz · 2025-03-13

**Overall Recommendation:** 3

**Summary:**

The paper proposes SOLD, a method for model-based reinforcement learning (MBRL) with object-centric world models. The paper is mainly a combination of ideas from OCVP object-centric video prediction and DreamerV3-style MBRL with latent dynamics model being shaped to be object-centric. The main motivation is to use object-centric representations as an inductive bias to improve sample efficiency and interpretability. For evaluation, the authors compare their methods with DreamerV3, TD-MPC2 and a variant of their architecture that excludes the object-centric representation. They build their own in environments on top of what seems to be Gymnasium-robotics environments to create object-centric manipulation tasks. Their approach proves to be equal or better to the baselines and also successful on some select benchmark tasks from DMC and MetaWorld.

## update after rebuttal

The authors addressed some of my concerns, but not all (missing MFRL baselines and standardized benchmarks). Nonetheless, their arguments and new evaluations are sufficient to raise my score.

**Claims And Evidence:**

The paper claims to be the first to do object-centric MBRL, which to the best of my knowledge is true.

In addition, the authors claim that object-centric representations are a good inductive bias to improve 1) interpretability and 2) sample efficiency of MBRL. The qualitative results shown by the authors in terms of slot rollouts do support the claim of improved interpretability. However, sample efficiency is hard to judge given that the authors only demonstrate this feature on environment that they made. It is then hard to say whether the environment was created in a way that favors the method or if the method is generally more sample efficient.

**Essential References Not Discussed:**

The related work section is missing some key works from the MBRL literature to name few:

[1] Deisenroth, Marc, and Carl E. Rasmussen. "PILCO: A model-based and data-efficient approach to policy search." Proceedings of the 28th International Conference on machine learning (ICML-11). 2011.
[2] Nagabandi, Anusha, et al. "Neural network dynamics for model-based deep reinforcement learning with model-free fine-tuning." 2018 IEEE international conference on robotics and automation (ICRA). IEEE, 2018.

**Experimental Designs Or Analyses:**

Experimental design lacks using standard benchmarks from the manipulation literature.

**Methods And Evaluation Criteria:**

The proposed method makes absolute sense. Object-centric representations are long-due for MBRL and are a great idea to improve interpretability, performance, and sample efficiency of RL methods.

The evaluation methods are not sufficient. The authors did a good job on the research direction and method implementation side of things. However, they fail to properly validate their proposed method. It is crucial to evaluate the proposed method on standard RL and manipulation benchmarks (e.g., MetaWorld, Robosuite, RLBench...). Most manipulation benchmarks require object-centric representations since manipulation by definition requires the robot to manipulate some objects in its environment.

The choice of baselines is adequate, though it is quite minimal (check my comment in the weakness section).

**Other Comments Or Suggestions:**

None

**Other Strengths And Weaknesses:**

**Strengths:**

- Using object-centric representation for MBRL is a great idea and a very promising direction.
- The paper is well-written an easy to follow
- The method includes multiple key design choices that enabled its performance on the target tasks.
- The proposed approach improves interpretability of MBRL, which is a major claim of the work that is well-supported by the experiments.
- The evaluation showcases the improvement in sample efficiency on the chosen tasks.

**Weaknesses:**

- My main concern is with the evaluation of the work. The authors should include some evaluation on more standard benchmarks that are also object-centric such as the ones from RLBench, MetaWorld, and Robosuite. The authors show the success of their methods on some select standardized benchmarks but they do  not compare their method to baselines in these domains.
- The paper lacks ablations of the key design choices proposed in the paper.
- The choice of baselines is minimal (sufficient to validate the claims, but not sufficient to highlight the benefits of the paper). I would suggest adding a baseline on model-free RL with the object-centric representations.
- While the method is interesting, the novelty of the work is quite limited by the fact that it is a mere combination of OCVP and DreamerV3

**Questions For Authors:**

- Given that author benchmark include object-centric tasks, what was the main motivation for building custon environments to evaluate the method? from an outsider's perspective and without you properly motivating this choice, it sounds like the environments were built to favor the proposed method.
- How would your method perform on environments with more complex action spaces? Since action-conditioning is one novelty from the side of object-centric video prediction, it would be interesting to see this working on more complex action spaces.

**Relation To Broader Scientific Literature:**

The paper is a great combination of OCVP and DreamerV3, bringing recent ideas from video prediction to model-based RL for visual environments.

**Theoretical Claims:**

No theoretical claims.

---

> ### Author Rebuttal · Authors · 2025-04-01
>
> We want to thank the reviewer for the detailed feedback. We are encouraged that they found our idea strong and clear, and that they appreciate the improvement in interpretability and sample efficiency. We aim to answer remaining questions and concerns below, but will incorporate all feedback for the paper (such as the discussion of key works from the model-based RL (MBRL) literature) into the final version.
>
> **Why are the evaluations performed on non-standard benchmarks/ custom environments?**
>
> We agree that comparative evaluation on diverse standard benchmarks, such as RLBench, is a crucial direction to explore. Our motivation for the selected evaluation is twofold: (1) feasibility with current OC representation (OCR) learning methods and (2) explicit investigation into relational reasoning capabilities.
>
> At the moment, many of these benchmarks are still beyond the capabilities of OC methods. The main purpose of our work is to evaluate the benefit OCRs can bring to learning control policies on environments where current methods can learn sufficiently disentangled representations to accurately investigate this question.
>
> Moreover, while we agree that all robotic manipulation is at its core object-centric, we are interested in specifically studying skills that go beyond the abilities of current methods operating on holistic representations (akin to prior work investigating OCRs in RL (https://arxiv.org/abs/2302.04419, https://arxiv.org/abs/2011.14381, https://arxiv.org/abs/2404.01220)). Improved relational reasoning, characterized as a cornerstone of human intelligence, is such a skill that is insufficiently challenged in standard benchmarks and that becomes increasingly important as RL agents are tasked to perform tasks based on high-level semantic instruction.
>
> **Did you perform ablations of key design choices?**
>
> We have added ablations to the key design choices here: https://anonymous.4open.science/r/sold-rebuttal/ABLATIONS.md
>
> **I would suggest adding a baseline on model-free RL with the OCRs.**
>
> We thank the reviewer for characterizing the comparative baselines we employed as adequate, but agree that adding an OC model-free (MF) RL method would add an interesting dimension to compare the performance of MB vs. MF methods operating on OCRs. Given the one-week timeframe for the rebuttal, implementing and evaluating such a baseline is beyond our means, and we have focused our efforts on the extensive ablations and additional experiments requested by the reviewers that were feasible within this time frame. We thank the reviewer for this suggestion and consider the comparison to OC-MFRL to be an interesting direction for future work.
>
> **Is the method ***"a mere combination of OCVP and DreamerV3"***?**
>
> We agree that extending OCVP to an action-conditional setting and designing an architecture to perform MBRL on the basis of this representation is central to our work. However, we believe that *“a mere combination of OCVP and DreamerV3”* undervalues the contribution. Both the action-conditional OC prediction and the MBRL on the basis of sequences of object slots in latent imagination (enabled by the SAT architecture) are novel contributions that address non-trivial challenges in OC-MBRL, which we believe to be of interest to the community.
>
> ***"How would your method perform in environments with more complex action spaces?"***
>
> We appreciate this question, since accurately modeling the effects of actions is crucial for learning control. While we have included predictions on DM-Control environments, where the action-spaces are substantially different from the position + gripper control of our task suite, we aim to show the feasibility of our world model to apply to more complex action-spaces. Therefore, we have performed additional experiments on:
> 1. The Sketchy dataset (https://arxiv.org/pdf/1909.12200), which features 7-dimensional actions, extending the action-space of our environments by a rotational component, and
> 2. A custom Moving Shapes environment featuring multiple shapes of distinct colors, where the action-space encodes information about both *what* happens (e.g., move upwards) and *to which* object (e.g., the red cube) and allows for actions to simultaneously apply to all objects.
>
> On the Sketchy dataset, results show that our prediction model is capable of accurately predicting the movement of the robot even under gripper rotations. For the Moving Shapes environment, we observe that the dynamics model is able to correctly associate actions given as a single vector to the different objects they apply to.
> We encourage you to view the result of this investigation here https://anonymous.4open.science/r/sold-rebuttal/COMPLEX_ACTION_SPACES.md
>
> ---
>
> We appreciate the reviewer's characterization of OCRs as long due for MBRL and his recognition of our idea. We hope that we have been able to address outstanding questions and concerns, and would be very grateful if you would consider updating your score.

---

> > ### Comment · Reviewer_o1iz · 2025-04-04
> >
> > The authors addressed some of my concerns, but not all (missing MFRL baselines and standardized benchmarks). Nonetheless, their arguments and new evaluations are sufficient to raise my score.

---

> > > ### Author Response · Authors · 2025-04-07
> > >
> > > We are glad that we were able to clarify questions and address some of the concerns with the additional evaluations.
> > >
> > > We agree that evaluating on standard benchmarks and more visually complex data is an important consideration. However, as noted also by other reviewers, it is currently challenging to evaluate across standard RL benchmarks, largely due to the limitations of slot attention and OCR learning methods in general.
> > >
> > > Despite this challenging setting, we believe there to be significant value in exploring the combination of RL and OCRs, as we are glad was noted by all reviewers, and believe SOLD to make an important contribution in this direction.
> > > Without meaning to claim that with the current OCR learning methods our approach is universally superior, we have been able to demonstrate the potential for both improved relational reasoning and interpretability, which we believe to be crucial capabilities for embodied agents.
> > >
> > > Exploring further how OCRs can generalize to arbitrary visual RL settings is a fascinating problem for future work, but we believe it to be beyond the self-contained insights that SOLD can bring to the underexplored research area of OC-MBRL.
> > >
> > > Finally, we would like to thank the reviewer for their insightful feedback both to more clearly position our current work and contributions, and outline crucial points for investigation in future work.

---

### Official Review · Reviewer_kT4P · 2025-03-14

**Overall Recommendation:** 3

**Summary:**

The work focuses on the development of a novel model-based reinforcement learning (RL) algorithm that utilizes an object-centered representation of visual scenes. The authors extend the standard model-based approach of Dreamer by incorporating a slot representation generated using the SAVi transformer model. Instead of relying on the traditional Dreamer recurrent state model (RSSM), the authors employ a newly developed object-centric dynamics model OCVP. One of the key innovations introduced by the authors is the utilization of a unique model for the reward function - the slot aggregation transformer (SAT). This model does not directly generate a scalar reward, but instead generates softmax logits of the softmax distribution from which the actual reward is sampled. Additionally, register tokens and ALiBi-based position encoding are employed. The authors conduct experiments in their custom environment involving a robotic arm and larger objects, comparing their approach with Dreamer and TD-MCP. They also indicate extension of their experiments to other platforms such as MetaWorld and simpler versions of DMControl.

## update after rebuttal
During the rebuttal phase, the authors conducted further experiments to evaluate the impact of SAT on the overall architecture and performed preliminary experiments on generalization. I agree with the authors that they have shown the promise of an object-centric approach that works at the level of monolithic approaches. However, this is not the first work in this field to demonstrate this. In general, I believe the proposed method has novelty, so I leave the current assessment Weak Accept.

**Claims And Evidence:**

The work is written in a clear and easy-to-understand language, although there are some issues with the notation and the introduction of the RL problem. The authors acknowledge some advantages of object-centric representations over monolithic ones, but they only discuss these advantages in a specific env. While it is possible to agree with some of the claims made by the authors, it is difficult to accept their assertion that this is the first approach based on an object-centric world model (see discussion of the literature in this area below).

**Essential References Not Discussed:**

The authors are encouraged to discuss and conduct an experimental comparison with the following approaches:
1)	Model-based - GOCA https://arxiv.org/abs/2310.17178
2)	Model-free – OCRL (cited as Yoon et al., 2023), OC-SA (https://arxiv.org/pdf/2208.03374)  и OCARL (https://arxiv.org/abs/2210.07802).

**Experimental Designs Or Analyses:**

The experiments conducted are not very convincing. Even in a setting designed specifically for their method, with large, clearly defined objects and without noise, the SOLD approach does not show a significant advantage over monolithic models in a standard setting, actually demonstrating slower learning. Additionally, the authors did not report the number of steps used to pre-train the OCVP model. There are only good results in a very specific context. Analysis of the results and comparisons with other methods in the MetaWorld and DM-Control environments were not provided. Another major drawback is the lack of comparison to other object-centric, model-free, and MBRL approaches (as described in the literature).

**Methods And Evaluation Criteria:**

The authors use a combination of existing models, including Dreamer, SAVi, and OCVP. However, the main new contribution of their work is the SAT reward model. For this, they have applied a number of novel techniques, resulting in a high-quality model.
One challenge they faced was using their own environment, as it is common in the research community to use more established benchmarks such as RoboSuite, CasualWorld, and Crafter. Additionally, the experiments with MetaWorld and DM-Control were not thoroughly conducted and were not discussed in detail.

**Other Comments Or Suggestions:**

The authors did not set the RL task itself, and part of the notation is not entered or deciphered, which makes it difficult to read the article. For example, the concepts of R_t, e_\eta, etc. are not introduced. Also, when describing the method, it is difficult to separate the authors' introductions and previously used techniques introduced back in DreamerV3.

**Other Strengths And Weaknesses:**

I am very impressed by the object-centric approach and agree with the authors that it is a promising area of research. However, there are currently some difficulties in applying it in interesting and acceptable environments within the RL community, largely due to limitations of slot attention and visual self-supervised learning methods in general. In this regard, it is important to mention the work done in this area and avoid expanding the range of specific environments. Instead, it would be better to focus on standard environments like CasualWorld or MetaWorld, which would allow for more reliable comparisons and evaluations. I would recommend that the authors conduct their experiments using these environments as a basis, using them as a demonstration and additional reference. Additionally, I would like to see evidence of the benefits of object-based learning, such as improved generalization and transferability to other tasks.

**Questions For Authors:**

1) How do the authors assess the possibilities of generalizing object-centric models, for example, to change colors?
2) How is the required amount of data estimated for OCVP pre–training?
3) I would like to see an ablation study on the SAT-related part of the model - how will it work with and without it?

**Relation To Broader Scientific Literature:**

The authors do not mention all the available work on object-centric representations in RL. Authors especially miss such works in the field of MBRL, for example, GOCA (see the literature section). At a minimum, they need to be discussed, and not to claim that such works do not exist.

**Theoretical Claims:**

The authors do not offer any theoretical analysis of the proposed model, but generally justify the loss functions used.

---

> ### Author Rebuttal · Authors · 2025-04-01
>
> We want to thank the reviewer for acknowledging that our approach, and the combination of object-centric (OC) representations and RL generally, is interesting and a promising research area. Further, we want to thank them for acknowledging the SAT architecture as one of the key novel contributions of our work. We address the raised questions below and will incorporate all the feedback for the paper and the provided references to related works into the final version.
>
> **Is this ***“the first approach based on an OC world model”***?**
>
> We want to thank the reviewer for raising this important point and providing a reference to the GOCA paper. While GOCA (called ROCA in the newest version) is specifically focused on value-based MBRL, using the world model to improve return estimates of the critic instead of learning through imagination inside world model rollouts, we want to make sure to characterize the novelty of our method accurately and not overlook any prior contributions. To this end, we add a more detailed discussion of the differentiation of our method to prior work on OC representations (OCRs) in RL (including all references provided by the reviewers) to the Appendix of our paper.
> Moreover, we would appreciate the reviewer’s feedback on the characterization of SOLD as the first method to learn purely inside imagined rollouts of an object-centric world model trained from pixel inputs.
>
> **Why are the evaluations performed on non-standard benchmarks/ custom environments?**
>
> To make the most of the limited space, we kindly refer to our reply to this question in the response to o1iz.
>
> **Why does SOLD not show a significant advantage over monolithic models in a standard setting?**
>
> We assume standard setting to refer to tasks that only consider distractor objects, without requiring explicit relational reasoning between those objects (Reach-, Push-, and Pick-Specific). While we want to point out that SOLD still outperforms state-of-the-art baseline methods on the more challenging Push and Pick instantiation of this task, TD-MPC2 performs best on the Reach-Specific environment.
> The experience used in pre-training to learn OCRs is a cost in the SOLD algorithm that is “paid” upfront and becomes valuable through improved representations over the course of training. On the easiest Reach-Specific task, TD-MPC2 has converged even before the experience required for pre-training SAVi is used up. To us, this is neither unexpected nor troubling, since the problem is too easy to master to warrant the extra effort in acquiring strong representations upfront. Specifically TD-MCP2, which operates in a reconstruction-free manner, can quickly extract the only relevant information from the visual representation, which in this case is the position of the green sphere and the robot’s end-effector.
> We still include this easier task to paint a fuller picture about when the use of OCRs has yielded a payoff for the total sample complexity and when baseline methods are already performing well.
>
> **How many frames are used to pre-train SAVi?**
>
> We have included this in the details about SAVi Pretraining in Section D.4 in the Appendix (Line 797). We pre-train SAVi for approximately 1 million frames (including the full episode in which the count of 1 million frames is reached) on all tasks, which we have found empirically to provide a sufficient basis to learn disentangled OC representations.
>
> ***"The code is missing, which makes it impossible to evaluate the correctness of the implementation and experiments."***
>
> The source code is available on the project website linked at the end of the abstract.
>
> **What do $R_t$, $e_\eta$, etc. refer to?**
>
> $R_t$ represents the return from time-step $t$. $e_\eta$ and $d_\eta$ represent a SAVi encoder and decoder model with parameters $\eta$, respectively. Due to the limited space for the main text, we have added explanations for $R_t$, $e_\eta$, and $d_\eta$ to the Notation in Section B of the Appendix.
>
> ***"I would like to see an ablation study on the SAT-related part of the model - how will it work with and without it?"***
>
> We have added ablations to key design choices of our method to better justify the decisions. The results of all experiments are available at https://anonymous.4open.science/r/sold-rebuttal/ABLATIONS.md
>
> **How do the authors assess the possibilities of generalizing OC models, for example, to change colors?**
>
> This is an interesting point for further investigation, and we have conducted a preliminary experiment changing the set of colors from originally 16 to 32 novel values (see https://anonymous.4open.science/r/sold-rebuttal/COLOR_CHANGE.md). The results indicate that both DreamerV3 and SOLD are able to generalize to this setting well and that stronger out-of-distribution experiments are required to inspect this property in a dedicated investigation.
>
> ---
>
> We hope our response has clarified your concerns, and would appreciate it if you considered updating your score.

---

> > ### Comment · Reviewer_kT4P · 2025-04-03
> >
> > I would like to express my gratitude to the authors for responding to my comments and questions. Additionally, I want to mention that the authors have conducted further experiments to evaluate the impact of SAT on the overall architecture and performed preliminary experiments on generalization. Overall, I appreciate the work, but the experimental results regarding monolithic versions may still not be sufficiently convincing to bring the community closer to understanding the usefulness of object-based representations.

---

> > > ### Author Response · Authors · 2025-04-07
> > >
> > > We are pleased that we were able to address the reviewers' comments, and that they appreciate the additional experiments we performed. We are glad that we have been able to clarify technical questions and we want to take this opportunity to comment on the outlook-oriented question of why we *do* believe that our work helps bring the community closer to understanding the utility of OC representations for control.
> > >
> > > While we do not claim that our learning method is universally superior to SoTA methods, we make important progress on *specific but crucial* properties, namely relational reasoning capabilities and interpretability of the agent's decision-making.
> > >
> > > While interpretability is not straightforwardly quantifiable, we are encouraged that all reviewers commented positively on this aspect of our work. Investigating attention weights to understand the workings of transformer models, while prominent in self-supervised CV and NPL research, is rarely considered in behavior learning. Visualizing which specific parts of a visual scene the agent considers relevant for decision making is, to our knowledge, a novel feature in RL, and we believe it adds an interesting dimension to the current MBRL landscape. Moreover, as OC representation learning methods improve and generalize to larger images of higher visual complexity, the ability to selectively attend to task-relevant information while ignoring other visual elements will only become more important.
> > >
> > > Relational reasoning skills, on the other hand, are explicitly measurable through problems like ranking (Specific-Relative) or odd-one-out (Distinct), and we aim to assess them through the introduced task suite (which we will also make publicly available). In this respect, we are able to improve the efficiency with which these problems can be learned. Specifically, SOLD with an overall success rate of 82% outperforms the second best method DreamerV3 (56.4% success rate) by just over 25%, which we do consider to be a significant improvement. This performance gap is achieved despite our choice to include simpler problems, such as Reach-Specific (to which we assume the reviewer’s comment *"the SOLD approach does not show a significant advantage over monolithic models in a standard setting, actually demonstrating slower learning"* refers).
> > > We would like to emphasize that this outcome is neither unexpected nor worrisome to us, as the entirety of our evaluation clearly illustrates how the benefits of learning OC representation becomes increasingly pronounced as task-complexity grows.
> > > We are convinced that these results, combined with the improved interpretability, are compelling in highlighting the utility and potential of OCRs for control.
> > >
> > > Finally, we would like to express our gratitude to the reviewer for providing detailed feedback on our work, including adding relevant related work, helping us to clarify the intent and claims of our work, and improving the clarity of the notation. We are encouraged that they highlight the potential of our approach and of OCRs for control in general.

---

### Decision · Program_Chairs · 2025-05-01

**Decision:**

Accept (poster)

**Comment:**

The paper proposes a model-based RL algorithm that learns object-centric dynamics models in an unsupervised manner from pixel inputs. The proposed method outperforms DreamerV3 and TD-MPC2 - state-of-the-art model-based RL algorithms - in benchmarks introduced in the paper. All reviewers appreciate the object-centric design of the dynamics model, but raise concerns regarding the simplicity of the environments considered.
They suggest that the paper should include evaluations on more standard benchmarks that are also object-centric such as the ones from RLBench, MetaWorld, and Robosuite.
However, since there is little research on object-centric dynamics models, all reviewers are positive for the acceptance of the paper.